# Fourier Amplitude and Correlation Loss: Beyond Using L2 Loss for Skillful Precipitation Nowcasting

**Chiu-Wai Yan**    **Shi Quan Foo**    **Van Hoan Trinh**    **Dit-Yan Yeung**
The Hong Kong University of Science and Technology
`{cwyan, sqfoo, vhtrinh}@connect.ust.hk`    `dyyeung@cse.ust.hk`

**Ka-Hing Wong**    **Wai-Kin Wong**
Hong Kong Observatory
`{khwong, wkwong}@hko.gov.hk`

## Abstract

Deep learning approaches have been widely adopted for precipitation nowcasting in recent years. Previous studies mainly focus on proposing new model architectures to improve pixel-wise metrics. However, they frequently result in blurry predictions which provide limited utility to forecasting operations. In this work, we propose a new Fourier Amplitude and Correlation Loss (FACL) which consists of two novel loss terms: Fourier Amplitude Loss (FAL) and Fourier Correlation Loss (FCL). FAL regularizes the Fourier amplitude of the model prediction and FCL complements the missing phase information. The two loss terms work together to replace the traditional $L_2$ losses such as MSE and weighted MSE for the spatiotemporal prediction problem on signal-based data. Our method is generic, parameter-free and efficient. Extensive experiments using one synthetic dataset and three radar echo datasets demonstrate that our method improves perceptual metrics and meteorology skill scores, with a small trade-off to pixel-wise accuracy and structural similarity. Moreover, to improve the error margin in meteorological skill scores such as Critical Success Index (CSI) and Fractions Skill Score (FSS), we propose and adopt the Regional Histogram Divergence (RHD), a distance metric that considers the patch-wise similarity between signal-based imagery patterns with tolerance to local transforms. Code is available at `https://github.com/argenycw/FACL`.

## 1 Introduction

Precipitation nowcasting refers to the task of predicting the rainfall intensity for the next few hours based on meteorological observations from remote sensing instruments such as weather radars, satellites and numerical weather prediction (NWP) models. The development of a precise precipitation nowcast algorithm is crucial to support weather forecasters and public safety, as it could facilitate timely alerts or warnings on severe precipitation and mitigate their impact on the community through early preventive actions. Sharp precipitation nowcast imagery that is perceptually similar to the actual observations (such as radar images) is equally important for weather forecasters to comprehend how the severity of precipitation will evolve in space and time, as well as to diagnose the rapid evolution of the underlying weather systems in real-time forecasting operations.

Besides the traditional optical-flow and NWP models, deep learning models have also been widely explored and adopted for precipitation nowcasting in recent years. The research community generally formulates the task as a spatiotemporal prediction problem, where a sequence of input radar or satellite maps is given, and the future sequence needs to be predicted or generated. Although multiple previous attempts proposed solid improvements to the model to grasp the spatiotemporal dynamics,

38th Conference on Neural Information Processing Systems (NeurIPS 2024).

deep learning models can result in blurry predictions in real-life datasets featuring precipitation patterns such as radar echo and satellite imagery. Consequently, they provide limited operational utility [1] in weather forecasts.

The blurry prediction in multiple deep learning models is believed to be caused by the use of pixel-wise losses such as the Mean Squared Error (MSE), which entangles the probability into model prediction. In other words, the uncertainty of the image transformation leads to obfuscation of the surrounding pixels in the prediction. Nevertheless, solely improving the model capability could not resolve this issue due to the high spatial randomness of the atmospheric dynamics. In order to suppress the ambiguity of the model output, an emerging approach is to utilize generative models such as generative adversarial networks (GANs) and diffusion models. In this paper, we introduce an alternative approach, which is to modify the loss function such that the model focuses on recovering the high-frequency patterns. By utilizing the Fourier domain, we would like to shed light on a deterministic, non-generative method that can sharpen the spatiotemporal predictions with negligible sacrifice to its correctness.

To achieve the desired sharpness, we propose the Fourier Amplitude Loss (FAL), a loss term that improves the prediction of high frequencies by regularizing the amplitude component in the Fourier space. Supported by empirical validation, we further propose the Fourier Correlation Loss (FCL), a complementary loss term that provides information on the overall image structure. Furthermore, we have developed a training mechanism that alternates between FAL and FCL based on an increasing probability of employing FAL throughout the training steps. We name this combined loss function the Fourier Amplitude and Correlation Loss (FACL). FACL is computationally efficient, parameter-free, and model-agnostic and it can be directly applied to a wide range of state-of-the-art deep neural networks and even generative models. Extensive experiments show that compared to MSE, FACL results in forecasts that are both more realistic and more *skillful* (i.e., high performance with respect to several meteorological skill scores). To the best of our knowledge, we are the first to substantially replace the spatial MSE loss with spectral losses without using generative components on the spatiotemporal prediction problem, demonstrating the novelty and significance of our approach.

Our main contributions are summarized as follows:

- We propose the Fourier Amplitude and Correlation Loss (FACL), which is constituted by sampling between the Fourier Amplitude Loss (FAL) for regularizing the spatial frequency of the predictions to enable clarity and sharpness, and the Fourier Correlation Loss (FCL), a modified loss term that is cohesive with FAL to capture the overall image structure.
- Theoretical and empirical studies show that FAL boosts the image sharpness significantly while FCL complements the missing information for accuracy.
- We apply FACL to replace the MSE reconstruction loss in generative models. Results show that generative models with FACL perform better with respect to most of the metrics.
- We propose the Regional Histogram Divergence (RHD), a quantitative metric to measure the distance between two signal-based imagery patterns with tolerance to deformations. RHD considers both the regional similarity and the visual likeness to the target.

## 2 Related Works

### 2.1 Precipitation Nowcasting as a Spatiotemporal Prediction Problem

Previous works generally formulate precipitation nowcasting as a spatiotemporal predictive learning problem. Given a sequence of observed tensors with length $t$: $X_1, X_2, ..., X_t$, the problem is to predict the future $k$ tensors formulated as follows:

$$\underset{X_{t+1},...,X_{t+k}}{\arg\max} \quad p(X_{t+1}, ..., X_{t+k} \mid X_1, X_2, ..., X_t) \tag{1}$$

Based on this formulation, numerous variations of convolutional RNN models were proposed to model both spatial and temporal relationships in the data. ConvLSTM [2] first proposed to integrate convolutional layers into LSTM cells, with the recurrence forming an encoder-forecaster architecture. PredRNN [3] replaced the ConvLSTM units with ST-LSTM units and modified the structure such that the hidden states flow in both spatial and temporal dimensions in a zigzag pattern. MIM [4] replaced the forget gate in ST-LSTM with another RNN unit, forming a memory-in-memory structure to learn

higher-order non-stationarity. Moreover, advanced modifications such as reversed scheduled sampling [5], gradient highway [6], etc. [7, 8] were proposed to further improve the overall performance of the model. With the breakthroughs brought by transformers and self-attention mechanism, space-time transformer-based models such as Rainformer [9] and Earthformer [10] were proposed to model complex and long-range dependencies.

On the other hand, CNN models have also been widely explored for the task as a video prediction problem. Inspired by the U-Net structure used in earlier works [11, 12], SimVP achieves remarkable performance and efficiency by adopting an encoder-translator-decoder structure with mostly convolutional operations. Among the parts, the translator (temporal module) is found to benefit from MetaFormer (an architecture with both token mixer and channel mixer) in subsequent studies [13, 14]. TAU [15] further demonstrated the effectiveness of the structure by adopting depthwise convolution followed by $1 \times 1$ convolution as the temporal module.

Conventional precipitation nowcasting tasks and video prediction tasks evaluate the output mainly with pixel-wise or structural metrics such as Mean Absolute Error (MAE), Mean Squared Error (MSE) and Structural Similarity (SSIM) Index. To better consider the hits and misses of signal-based reflectivity, the Critical Success Index (CSI; equivalent to Intersection over Union, IoU), Fractions Skill Score (FSS) and Heidke Skill Score (HSS) belong to another type of metrics widely used in meteorology. To distinguish these scores from those used in the traditional machine learning literature, we refer to this metric type as *skill scores* in the remaining sections of the paper.

## 2.2   A Non-deterministic Perspective on Atmospheric Instability

Traditional models can result in blurry predictions at longer lead times, causing difficulty in forecasting operations. To address it, recent works leverage generative models such as GANs and diffusion models to promote realistic forecasts which could bring more insightful observation to forecasting operations. DGMR [1] utilizes a GAN framework with discriminators in both the spatial and temporal dimensions to ensure that the predicted images are sufficiently realistic and cohesive. LDCast [16] uses latent diffusion to generate a diverse set of outputs for ensemble forecasting. Meanwhile, the literature in video generation strives to generate realistic output frames with generative models. PreDiff [17] introduces a knowledge alignment mechanism with domain-specific constraints while adopting a latent diffuser for quality forecasts. DiffCast [18] appends a diffusion component as an auxiliary module to improve the realisticity of the forecasts. It is worth mentioning that the literature in video generation [1, 19, 20, 21, 22] also exhibits potential in generating high-quality nowcastings despite not specifically being designed to handle precipitation. Unlike works in video prediction, instead of evaluating the output quality with pixel-wise similarity, perceptual metrics such as LPIPS [23] and Fréchet Video Distance (FVD) [24] are predominantly used.

These works usually formulate the task as an unsupervised or semi-supervised learning problem with the results being non-deterministic based on a random prior, enabling the possibility of ensemble prediction. However, studying each prediction individually is less reliable as the prediction is unexplainably affected by the random prior. Furthermore, the inference efficiency of the diffusion model is poor due to the iterative nature of the reverse diffusion sampling process. Concerning the drawbacks of generative models, our method is proposed to be efficient, deterministic, and accurate at both the pixel and perceptual levels, bridging the advantages of both probabilistic video prediction and non-deterministic video generation.

## 2.3   Supervised Learning Problems That Utilize Fourier Transform

Spectral analysis in the Fourier space is a common practice for DNNs to study the features in terms of frequency. Rahaman et al. [25] proposed a property known as the spectral bias, which causes DNN models to be biased towards low-frequency functions. A follow-up study [26] theoretically showed that DNN models have a much slower convergence rate toward high-frequency components. Such observations motivate subsequent works to apply Fourier-based loss terms extensively in tasks such as super-resolution (SR) where fine details are crucial.

Despite the existence of works that apply Fourier transform amid the model feed-forward pipeline [27, 28, 29, 30, 31], here we focus on works that utilize spectral transform in the loss function or as a regularization term. Inspired by the JPEG compression mechanism, the Frequency Domain Perceptual Loss [32] compares the Discrete Cosine Transform (DCT) of the model

output in 8×8 non-overlapping patches: $L(y, \hat{y}) = c \odot \|\mathrm{DCT}(y) - \mathrm{DCT}(\hat{y})\|_2^2$, where $c$ is a vector of constants computed from the quantization table and training set. The Focal Frequency Loss [33] compares the element-wise weighted Fast Fourier Transform (FFT) output: $L_{\mathrm{FFL}} = \frac{1}{MN} \sum_{u=0}^{M-1} \sum_{v=0}^{N-1} w(u, v) |\mathrm{FFT}(y)_{u,v} - \mathrm{FFT}(\hat{y})_{u,v}|^2$, where $w(\cdot, \cdot)$ is a dynamic weight matrix and $|\cdot|$ refers to the absolute operator on complex numbers. Moreover, the Fourier Space Loss [34] decomposes the Fourier output (in complex) into amplitude and phase and measures their difference separately as a GAN loss component.

Although these losses were proposed specifically for the SR task, we find the problem setting similar to spatiotemporal forecasting in terms of the requirement for high-frequency fine details and the involvement of a ground-truth label. While taking advantage of the Fourier space as a spectral analysis is intuitive, choosing the proper distance metrics and additional weighting is tricky. This motivates us to propose a new loss function with consideration of the spectral property on the spatiotemporal forecasting problem.

# 3 Our Methods

In this section, we start by arguing why a naive implementation of the Fourier loss does not benefit the model compared with the MSE loss in the image space. Then, we will discuss the motivation and details of our proposed FACL.

## 3.1 Preliminaries

An image $X$ can be interpreted as a 2D matrix with the transformed Fourier series, $F$. The orthonormalized Discrete Fourier Transform (DFT) output and its corresponding inverse Discrete Fourier Transform are formulated as:

$$F_{pq} = \frac{1}{\sqrt{MN}} \sum_{m=0}^{M-1} \sum_{n=0}^{N-1} X_{mn} e^{-i2\pi(\frac{mp}{M} + \frac{nq}{N})}; \; X_{mn} = \frac{1}{\sqrt{MN}} \sum_{p=0}^{M-1} \sum_{q=0}^{N-1} F_{pq} e^{i2\pi(\frac{mp}{M} + \frac{nq}{N})} \quad (2)$$

where $M$ and $N$ are the height and width, respectively, of the image $X$.

To constrain model convergence via the spatial frequency components of its prediction, one naive design is to regularize the $L_2$ norm of the displacement vector between the ground truth and prediction in the Fourier space apart from the image space. Parseval's Theorem shows that such design is linearly proportional to the spatial MSE loss, and the detailed proof can be found in Appendix B.

Since this straightforward regularization does not differ from the MSE loss in the image space, the common adaptations from previous works are either to apply weighting on different frequencies or to decompose the Fourier features into amplitude $|F|$ and phase $\theta_F$ with the following definitions:

$$|F| = \sqrt{F_{\mathrm{real}}^2 + F_{\mathrm{imag}}^2}; \; \theta_F = \arctan(\frac{F_{\mathrm{real}}}{F_{\mathrm{imag}}}), \quad (3)$$

where $F_{\mathrm{real}}$ and $F_{\mathrm{imag}}$ are the real and imaginary parts, respectively, of the complex Fourier vector $F$.

## 3.2 Fourier Amplitude Loss (FAL)

As the spectral bias indicates the lack of attention to the high-frequency components, we encourage the model to consider high-frequency patterns by applying a loss on the amplitude of each frequency band. Similar to previous works, we first apply DFT to obtain the spectral information. Using Equation (3), we extract only the Fourier amplitudes ($|F|$) in the Fourier space and compare them in $L_2$:

$$\mathrm{FAL}(X, \hat{X}) = \frac{1}{MN} \sum_{p=0}^{M-1} \sum_{q=0}^{N-1} (|F|_{pq} - |\hat{F}|_{pq})^2 \quad (4)$$

where $F$ is the DFT output of $X$ as formulated in Eq. (2). Note that the formulation is subtly different from minimizing the $L_2$ norm of the displacement vector that prediction deviates from ground truth in the Fourier domain. The new formulation based on the Fourier amplitude of the images only is invariant to global translation. This reduces the spatial constraint induced by MAE and MSE losses. A detailed analysis can be found in Appendix D.

Despite retaining the high-frequencies by dropping the Fourier phase, FAL alone is insufficient to reconstruct the image. As $X \mapsto |F|$ is a many-to-one mapping, there exist multiple $X$ to have the same Fourier amplitude matrix. Thus, simply minimizing Eq. (4) can likely converge to an undesirable critical point. A high-level interpretation is that only image sharpness is retained by this loss while the information regarding the actual shape and position is lost with the Fourier phase discarded. Hence, on top of FAL, we require another loss term to compensate for the missing information, leading to our upcoming proposal of the FCL term. An alternative perspective via the mathematical formulation can be found in Appendix C.

### 3.3 Fourier Correlation Loss (FCL)

To remedy the missing information resulting from FAL, there are several approaches to take the image structure into account. A straightforward way is minimizing the difference of the Fourier phase between the prediction and the label, but it fails as $\theta_F$ obtained under DFT is discontinuous. Another approach is to compute the cosine distance in the Fourier domain without extracting $\theta_F$ directly. However, our preliminary experiments reveal that such formulation is unstable in reconstructing the image structure. Ultimately, we propose to implement the correlation between the generated output and ground truth in the Fourier domain and adopt it as the Fourier Correlation Loss (FCL) in our proposed loss:

$$\text{FCL}(X, \hat{X}) = 1 - \frac{\frac{1}{2} \sum [F\hat{F}^* + \hat{F}F^*]}{\sqrt{\sum |F|^2 \sum |\hat{F}|^2}}, \tag{5}$$

where $\sum$ here is a shorthand for the summation over all elements of the Fourier features and $*$ denotes the complex conjugate of the vector. FCL plays a significant role during training as it is responsible for learning the proper image structure while FAL can be treated as a regularization to promote the high-frequency components that FCL fails to capture.

The formulation of FCL has a similar format to the Fourier Ring Correlation (FRC) and Fourier Shell Correlation (FSC) widely used in image restoration and super-resolution of cryo-electron microscopy [35, 36, 37, 38, 39, 40]. However, both FRC and FSC pre-define a specific region of interest (either a ring or a shell) on the Fourier features. In contrast, we extend the region of interest to the entire map, considering the global spectral bands with all frequencies. To ensure the score is real and commutative, we take the average of $F\hat{F}^*$ and $\hat{F}F^*$ in the numerator. The denominator performs normalization such that FCL only focuses on the image structure in the global view rather than the absolute brightness. Unlike FRC (without $1-$), FCL spans the range [0, 2], where larger values refer to a negative correlation and smaller values refer to a positive correlation. Further analysis of FCL from the gradient aspect can be found in Appendix E.

### 3.4 Proposed Approach: Random Selection between FAL and FCL

While it is straightforward to apply the overall loss function as a linear combination of FAL and FCL, we find it tricky to determine the weighting of the components in our preliminary studies. Instead, we offer a more controllable solution – to alternate FAL and FCL as shown below:

$$\text{FACL}(X, \hat{X}, t) = \begin{cases} \text{FAL}(X, \hat{X}), \text{ if } p > P(t) \\ \text{FCL}(X, \hat{X}), \text{ otherwise} \end{cases} \tag{6}$$

where $p$ is sampled randomly and uniformly in [0, 1] and $P(t)$ is a pre-defined threshold decreasing during the training process as shown in Figure 1. $P(t)$ always decreases from 1 to 0 such that the model is first trained with $100\%$ FCL that takes image structure into account, and then the models are more frequently trained with FAL which improves the image sharpness.

Since FCL loses information on the overall brightness, the model could not achieve proper brightness at the early stage where FCL dominates the learning objective. To address it, we append a sigmoid function in the output layer of the model. This constrains the model output in the range [0, 1] to prevent the model from converging to a sub-optimal state with an undesirable range of output values.

Overall, the following modifications are applied to the models:

- Training loss function of the models involving FAL and FCL is formulated in Eq. (6).

- A sigmoid layer is appended to the end of the model. For RNN models, the sigmoid function is applied before the output of the last RNN stack.
- To coordinate with the decreasing threshold, the cosine annealing learning rate scheduler is used rather than the conventional reduce-on-plateau scheduler.

### 3.5 A New Metric: Regional Histogram Divergence (RHD)

Previous works in video prediction tend to use pixel-wise metrics such as MSE and MAE to measure the difference between the prediction and labels. Such a choice of metrics might not fit spatiotemporal data for two reasons: (1) reasonable pixel shifts are highly penalized, and (2) the overall distribution of values is ignored. This encourages the models to output blurry predictions while regional uncertainty diffuses outward over time. By inverse, deep perceptual metrics such as LPIPS, Inception Score (IS) and Fréchet Video Distance (FVD) suffer from the knowledge bias between multi-channel pictures (as pre-trained on ImageNet) and monotonic signal-based intensities.

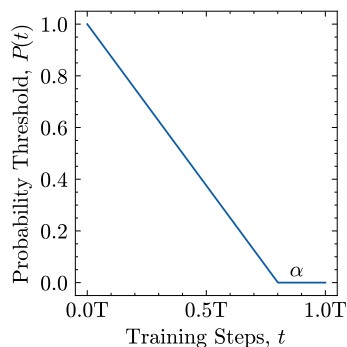

One of the metrics that consider both the previous two factors is the Fractional Skill Score (FSS), which is widely used in meteorology. After splitting the image into $N_x \times N_y$ smaller patches, where $N_x$ and $N_y$ control the shift of precipitation events we tolerate, we obtain the FSS score as follows:

$$\text{FSS} = 1 - \frac{\sum_{i=1}^{N_x} \sum_{j=1}^{N_y} (F_{i,j} - O_{i,j})^2}{\sum_{i=1}^{N_x} \sum_{j=1}^{N_y} F_{i,j}^2 + \sum_{i=1}^{N_x} \sum_{j=1}^{N_y} O_{i,j}^2}, \quad (7)$$

Figure 1: The pre-defined probability threshold function $P(t)$ over training steps $t$ with $T$ total steps. $\alpha$ determines the ratio of the training steps where $P(t) = 0$.

where $F_{i,j}$ and $O_{i,j}$ refer to the fraction of predicted positives and fraction of observed positives, respectively, of the patch in the $i$-th row and $j$-th column. Based on this formulation, the intensities are free to reposition within the patch window, granting tolerance to translation and deformation. Nevertheless, one drawback of FSS is that the pixel range is only categorized into two classes: positives and negatives. For a threshold of $0.5$, a pixel value of $0$ is treated the same as a pixel value of $0.49$, resulting in a huge error when viewing the per-patch precision. This means that the choice of threshold induces a bias in evaluating the forecasting performance of models.

To improve the representation, we propose the Regional Histogram Divergence (RHD), a variation of FSS that exhibits smaller errors within a class. Instead of categorizing the pixel values into 'hits' and 'misses', we divide the values into $n$ bins and count the frequency of each bin, obtaining a histogram for each patch. Next, we compare the average Kullback–Leibler (KL) divergence on the histograms. Mathematically, the RHD between two sets of bins can be expressed as:

$$\text{RHD} = \frac{1}{N_x N_y} \sum_{i=1}^{N_x} \sum_{j=1}^{N_y} D_{\text{KL}}(O'_{i,j} || F'_{i,j}) = \frac{1}{N_x N_y} \sum_{i=1}^{N_x} \sum_{j=1}^{N_y} \sum_{x \in \mathcal{X}} O'_{i,j}(x) \log \frac{O'_{i,j}(x)}{F'_{i,j}(x)},$$

where $F'_{i,j}$ and $O'_{i,j}$ correspond to the predicted and observed discrete probability distributions, respectively, among the set of bins $\mathcal{X}$ of the patch in the $i$-th row and $j$-th column.

Different from FSS where the proportions of positives are subtracted directly, RHD instead compares the distributional difference in the context of the histograms. This not only increases the precision of each class/bin, but also heavily penalizes blurs since blurring forms a Gaussian-like distribution in the histograms while sharp intensities should have an 'M-shape' bimodal distribution. If the patch-wise distribution of the two images is identical, the corresponding RHD is 0. The larger the RHD is, the more different the two sets of patches behave. Furthermore, RHD is formulated to be a mean KL divergence so it is always non-negative.

For simplicity and consistency, we choose the number of bins to be 10, divided uniformly within the range $[0, 1]$ for all datasets in our experiments. In real-life applications, non-uniform division can be applied for data in non-linear scales such as radar echo (in dBZ) to highlight specific ranges of values. When we compute the histograms, as 0 dominates in the imagery, we apply a threshold $\epsilon = 10^{-5}$ to exclude all small intensities.

## 4 Experiments

### 4.1 Experimental Settings

We evaluate the performance of our proposed method on a synthetic dataset and three radar echo datasets, namely, Stochastic Moving-MNIST, SEVIR [41], MeteoNet [42] and HKO-7 [43]. A more detailed description for each dataset can be found in Appendix A. To show that our method is effective and generic, we selected ConvLSTM [2] and PredRNN [3] (reported in Appendix I), two RNN-based models with different recurrence paths; SimVP [44], a CNN-based model and Earthformer [10], a transformer-based model. We trained the models with two variants: conventional MSE and FACL (as formulated in Eq. (6)). To compare with generative models as references, we also report the results of LDCast [16] (latent diffusion) and MCVD [21] (denoising diffusion) for all datasets. For Stochastic Moving-MNIST, we report two more models, i.e., PreDiff [17] (latent diffusion) and STRPM [45] (GAN-based). Appendix M reports the detailed setup and hyper-parameters.

In the upcoming sections, we will first present the setup and experimental results on the Stochastic Moving-MNIST dataset. After that, we will test the models with three real-world radar echo datasets. Extra studies on our methods are reported in the Appendix. Specifically, we report the ablation study of FAL, FCL and $\alpha$ in Appendix F, the running time of FACL in Appendix G, experiments on additional datasets in Appendix H, comparison with other potential loss functions in Appendix J, the performance when applying FACL to generative models in Appendix K, and characteristic analysis of RHD in Appendix L. To demonstrate the advantages of our method against counterparts for precipitation nowcasting, video prediction and video generation, we evaluate the models with a union of metrics from the areas. Specifically, we report the MAE and SSIM to show the pixel-wise and structural accuracy; LPIPS and FVD to show the deep perceptual similarity to ground truth; FSS and RHD to measure the similarity of the intensity distribution in different regions. For radar echo datasets, we further include the CSI and pooled CSI to reveal the models' capability of identifying potential extreme weather. Such a combination of metrics is believed to facilitate a comprehensive understanding of the pros and cons of the current state-of-the-art in precipitation nowcasting.

### 4.2 A Stochastic Modification of Moving-MNIST

The Moving-MNIST dataset has been a common benchmark to evaluate how well a model could predict motion in spatial preservation and temporal extrapolation. However, the nature of the Moving-MNIST is highly deterministic, which does not resemble the chaotic nature of the atmospheric system. Previous adaptations attempted to simulate the physical dynamics by introducing a set of complex motions such as rotation and scaling [10] or by applying an external force on collision [46]. We argue that the fundamental reason causing the blur in precipitation nowcasting is the intrinsic stochasticity of the motion caused by external factors unseen in the weather dataset, such as orographic effects, vertical wind shear, interaction with other weather systems, etc. Trained with such stochasticity, regular models with pixel-wise loss could consistently fail to provide quality prediction in the future lead time.

To verify our claim, we introduce a simplistic modification to the Moving-MNIST dataset. The standard Moving-MNIST dataset contains handwritten digits sampled from the MNIST dataset moving and bouncing with a constant velocity $(u_0, v_0)$ on the $64 \times 64$ image plane. To introduce stochasticity, we perturb the velocity with a random Gaussian noise $\epsilon$ at each time step. Details of the perturbation are shown in Appendix A. In the upcoming sections, we dub this dataset Stochastic Moving-MNIST and apply the experimental setting to this synthetic dataset. The performance of combinations of different losses and models can be found in Table 1 and qualitative visualizations of the corresponding methods are shown in Figure 2 and Appendix N. Note that the Stochastic Moving-MNIST is used in both training and evaluation to ensure that the models are well exposed to motion randomness.

In Table 1, our modification drastically improves the sharpness and realisticity for all tested models, as reflected by the vast reduction in LPIPS and RHD. In particular, FACL reduces up to 57% of LPIPS and 71% of RHD for the ConvLSTM model. The pixel-wise and structural metrics between the two losses are comparable. On the other hand, generative models result in much poorer MAE and SSIM, with skill scores like FSS still being worse than most of the baseline models. In Figure 2, we can observe that the model trained with MSE cannot reconstruct a clear spatial pattern, especially in the subsequent frames, while the model trained with FACL yields much sharper and higher quality

Table 1: Comparison of the quantitative performance of different losses for models trained on the Stochastic Moving-MNIST. The better score between MSE and FACL is highlighted in bold.

| Type | Model | Loss | Pixel-wise/Structural | | Perceptual | | Skill | Proposed |
| | | | MAE↓ | SSIM↑ | LPIPS↓ | FVD↓ | FSS↑ | RHD↓ |
|---|---|---|---|---|---|---|---|---|
| Pred. | ConvLSTM | MSE | 196.4 | 0.6975 | 0.2538 | 451.5 | 0.6148 | 1.1504 |
| | | FACL | **180.1** | **0.7463** | **0.1092** | **82.3** | **0.8172** | **0.3391** |
| | PredRNN | MSE | 173.8 | 0.7566 | 0.1875 | 337.8 | 0.7443 | 0.9559 |
| | | FACL | **162.1** | **0.7812** | **0.0869** | **63.3** | **0.8549** | **0.3000** |
| | SimVP | MSE | **175.5** | **0.7547** | 0.1943 | 350.6 | 0.7275 | 0.9819 |
| | | FACL | 180.2 | 0.7394 | **0.1335** | **211.9** | **0.8168** | **0.3579** |
| | Earthformer | MSE | 171.5 | 0.7641 | 0.1828 | 320.3 | 0.7532 | 0.9407 |
| | | FACL | **167.6** | **0.7768** | **0.0890** | **64.6** | **0.8463** | **0.3230** |
| Gen. | LDCast* | - | 234.0 | 0.7053 | 0.1541 | 110.7 | 0.6645 | 0.4343 |
| | MCVD | - | 219.8 | 0.7125 | 0.1033 | 44.7 | 0.7184 | 0.3941 |
| | STRPM | - | 154.0 | 0.7912 | 0.1017 | 117.4 | 0.8337 | 0.3216 |
| | PreDiff | - | 190.2 | 0.7570 | 0.0709 | 30.8 | 0.7975 | 0.3052 |

* The experiment setting for LDCast is changed to 8-in-8-out due to model constraints.

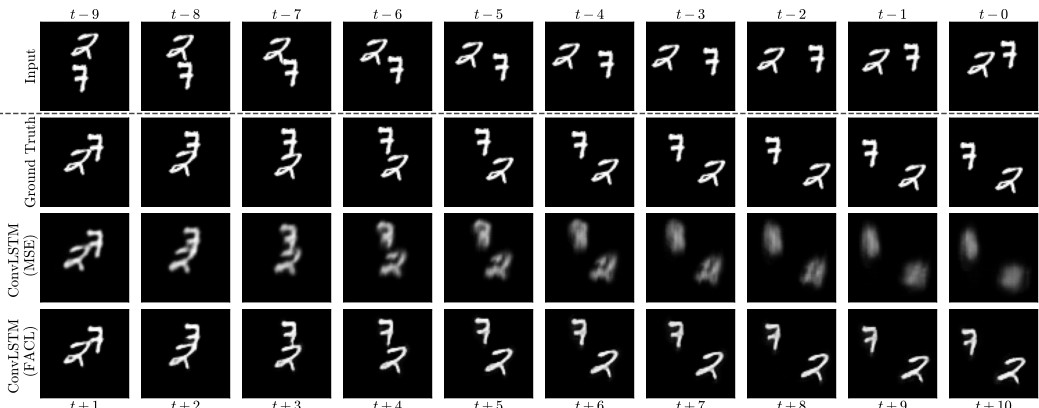

Figure 2: Output frames of the ConvLSTM model trained with different losses on Stochastic Moving-MNIST. From top to bottom: Input, Ground Truth, MSE, FACL.

outputs. Consequently, we can conclude that in this setting with the synthetic Stochastic Moving-MNIST, FACL demonstrates a significant improvement in perceptual metrics and skill scores, with the quality on par with that of generative models.

### 4.3 Performance on Radar-based Datasets

In this section, we extend the previous setup to general radar-based datasets. Apart from the distance metrics used in the last section, we further report the CSI with different pooling sizes. Following the previous works [43, 10], we measure multiple CSI scores with different thresholds ({16, 74, 133, 160, 181, 219} for SEVIR, {12, 18, 24, 32} for MeteoNet and {84, 117, 140, 158, 185} for HKO-7). The visualizations can be found in Figure 3 and more in Appendix N.

The results of Table 2 are similar to the observations in Table 1. Compared with the MSE baselines, FACL always improves the perceptual and skill scores. For sharp forecasts, the pooled CSI increases with the pooling size while for blurry forecasts, CSI shows no apparent difference based on pooling size. For some models, we observe a tiny decay in pixel-wise and structural metrics. For example, the Earthformer model trained with FACL on SEVIR has a $6.4\%$ increase in MAE, which is believed to be a natural trade-off since pixel-wise metrics have no tolerance for spatial transformation. Despite poorer pixel-wise performance, the perceptual metrics and skill scores always improve, as further illustrated by Figure 3 that only FACL predicts fine-grained extreme values. Regarding those generative models, although they could perform the best in deep perceptual scores like LPIPS and FVD, we still observe that they usually result in poorer skill scores. Moreover, it is noteworthy that FACL does not add any new parameters to the model. The change in the metrics solely indicates that FACL induces a shift of focus from pixel-wise accuracy to image quality and prediction skillfulness.

Table 2: Comparison of the quantitative performance of different losses for models trained on SEVIR, MeteoNet and HKO-7. MAE metrics is in the scale of $10^{-3}$. The better score between MSE and FACL is highlighted in bold.

| Type | Model | Loss | Pixelwise/Structural MAE↓ | SSIM↑ | Perceptral LPIPS↓ | FVD↓ | Skill CSI-m↑ | $CSI_4$-m↑ | $CSI_{16}$-m↑ | FSS↑ | Proposed RHD↓ |
|---|---|---|---|---|---|---|---|---|---|---|---|
| **SEVIR** | | | | | | | | | | | |
| Pred. | ConvLSTM | MSE | **26.35** | **0.7730** | 0.3683 | 510.2 | 0.3957 | 0.3965 | 0.4082 | 0.5252 | 1.5123 |
| | | FACL | 27.60 | 0.7624 | **0.3508** | **289.5** | **0.3984** | **0.4295** | **0.5073** | **0.5640** | **1.2087** |
| | Earthformer | MSE | **26.39** | **0.7701** | 0.3831 | 947.1 | **0.3999** | 0.3961 | 0.3976 | **0.5316** | 1.5363 |
| | | FACL | 28.09 | 0.7627 | **0.3575** | **384.3** | 0.3982 | **0.4129** | **0.4742** | 0.5244 | **1.3736** |
| | SimVP | MSE | **26.26** | **0.7643** | 0.3767 | 555.9 | 0.3989 | 0.3939 | 0.3956 | 0.5225 | 1.5244 |
| | | FACL | 27.55 | 0.7551 | **0.3476** | **243.8** | **0.4100** | **0.4387** | **0.5176** | **0.5656** | **1.1719** |
| Gen. | LDCast | - | 40.93 | 0.6647 | 0.3800 | 163.7 | 0.3000 | 0.3357 | 0.4411 | 0.3971 | 1.5988 |
| | MCVD | - | 32.88 | 0.7386 | 0.3239 | 99.6 | 0.3636 | 0.3981 | 0.5017 | 0.5230 | 1.3687 |
| **MeteoNet** | | | | | | | | | | | |
| Pred. | ConvLSTM | MSE | **6.47** | 0.9155 | 0.1504 | 247.9 | **0.4388** | 0.3989 | 0.3904 | 0.5036 | 0.2707 |
| | | FACL | 6.83 | **0.9170** | **0.1252** | **122.3** | 0.4161 | **0.4876** | **0.6041** | **0.5196** | **0.1944** |
| | Earthformer | MSE | **7.13** | **0.9045** | 0.1708 | 370.6 | **0.4004** | 0.3327 | 0.2946 | 0.4629 | 0.3213 |
| | | FACL | 8.01 | 0.9044 | **0.1589** | **203.2** | 0.3594 | **0.4038** | **0.5250** | **0.4833** | **0.2773** |
| | SimVP | MSE | **6.66** | **0.9128** | 0.1571 | 268.9 | **0.4221** | 0.3748 | 0.3627 | **0.4974** | 0.2820 |
| | | FACL | 7.21 | 0.9088 | **0.1450** | **128.4** | 0.4008 | **0.4513** | **0.5722** | 0.3826 | **0.2170** |
| Gen. | LDCast | - | 19.94 | 0.7295 | 0.3263 | 486.6 | 0.2353 | 0.3188 | 0.4804 | 0.1333 | 0.5594 |
| | MCVD | - | 13.18 | 0.8395 | 0.1549 | 55.7 | 0.3645 | 0.4559 | 0.6148 | 0.3838 | 0.2979 |
| **HKO-7** | | | | | | | | | | | |
| Pred. | ConvLSTM | MSE | 30.43 | 0.6664 | 0.3057 | 791.3 | 0.2772 | 0.2282 | 0.1702 | 0.2653 | 1.2453 |
| | | FACL | **29.72** | **0.7168** | **0.2962** | **569.1** | **0.3054** | **0.3040** | **0.3351** | **0.4045** | **0.7916** |
| | Earthformer | MSE | **31.62** | **0.6617** | **0.3186** | 939.1 | 0.2492 | 0.1976 | 0.1402 | 0.2367 | 1.3426 |
| | | FACL | 34.59 | 0.6004 | 0.3247 | **619.3** | **0.2812** | **0.2746** | **0.2962** | **0.3538** | **1.0752** |
| | SimVP | MSE | **30.93** | 0.6585 | 0.3039 | 808.1 | 0.2739 | 0.2227 | 0.1642 | 0.2617 | 1.2623 |
| | | FACL | 31.65 | **0.6803** | **0.2912** | **555.1** | **0.3018** | **0.3067** | **0.3223** | **0.3973** | **0.8660** |
| Gen. | LDCast* | - | 47.57 | 0.7269 | 0.3168 | 257.2 | 0.1846 | 0.2229 | 0.2486 | 0.3116 | 1.5542 |
| | MCVD* | - | 47.26 | 0.6813 | 0.3334 | 215.3 | 0.2576 | 0.2951 | 0.3233 | 0.4289 | 1.5836 |

\* HKO-7 data trained on LDCast and MCVD are down-scaled to 256 and 128 respectively and reshaped back to 480 for evaluation.

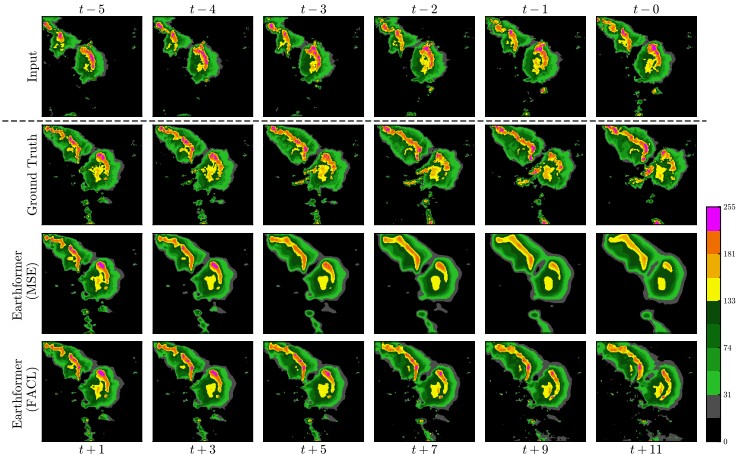

Figure 3: Output frames of the Earthformer models trained with MSE and FACL on SEVIR.

## 5   Conclusion

In this paper, we proposed the Fourier Amplitude and Correlation Loss (FACL). The two loss terms, Amplitude Loss (FAL) and Fourier Correlation Loss (FCL), encourage the model to focus on the Fourier frequencies and image structure correspondingly. Besides, we proposed a new metric, Regional Histogram Divergence (RHD), to measure the patch-wise similarity between two spatiotemporal patterns. We widely tested our methods on a synthetic dataset and three more real-life radar echo datasets, measured by metrics considering accuracy, realisticity and skillfulness. Extensive experiments reflect that our method yields sharper, more realistic and skillful forecasts with limited degradation in pixel-wise similarity.

Despite the remarkable performance of the FACL loss, our methods still have room for improvement. First, we assumed the data to be monotonic radar echo, which might not generalize well to multi-modal datasets featured in medium-range forecasts. Besides, our loss provides no regularization on temporal consistency, which may lead to the misalignment of temporal features between frames. The solution to these issues, however, will be open for future work.

## Acknowledgments and Disclosure of Funding

This work has been made possible by a Research Impact Fund project (R6003-21) and an Innovation and Technology Fund project (ITS/004/21FP) funded by the Hong Kong Government.

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

# A Details of the Datasets

**Stochastic Moving-MNIST.** Based on the same method generating vanilla Moving-MNIST, we further update the velocity of the digits at time $t$:

$$\begin{cases} u_t \leftarrow u_0 + \epsilon \\ v_t \leftarrow v_0 + \epsilon \end{cases} \text{, where } \epsilon \sim \mathcal{N}(0, 1) \tag{8}$$

Each unit corresponds to a pixel in the $64 \times 64$ image. Note that the expected trajectory is unchanged under this modification. However, the biased trajectory is exposed to the model as a stochastic factor influencing model training. Due to such behavior, models trained with the MSE loss are expected to exhibit a motion blur pattern along the direction of motion.

**N-body-MNIST.** To forge the chaotic nature of the Earth system, N-body-MNIST [10] was proposed to study the effectiveness of deep learning models. Rather than linear translation as in the conventional Moving-MNIST, digits in N-body-MNIST follow the N-body motion pattern, exerting an attractive force between digits and causing each other to circulate. Following the default setting of the original paper, the frame size is set to be $64 \times 64$ with $N = 3$. We use the same training, validation and test sets which contain 20000, 1000 and 1000 sequences respectively provided by the official repository.

**SEVIR.** The SEVIR dataset [41] is a spatiotemporally aligned dataset containing over 10,000 weather events in a $384\text{km} \times 384\text{km}$ region in the US spanning a period of four hours from 2017 to 2019. Among the five channels provided, we extract the NEXRAD Vertically Integrated Liquid (VIL) data product for precipitation nowcasting. Following previous works [10, 47], we predict the future VIL up to 60 minutes (12 frames) from 65 minutes of input frames (13 frames). We sample the test set from June 2019 to December 2019, leaving the remaining as the training set.

**MeteoNet.** MeteoNet [42] is an open meteorological dataset containing satellite and radar imagery in France. The data covers two geographic areas of $550\text{km} \times 550\text{km}$ on the Mediterranean and Brittany coasts, respectively, from 2016 to 2018. The time interval between consecutive frames is 5 minutes. Since there are missing values labeled as $-1$ in the raw rectangular data in shape $(565, 784)$, we preprocess it by filling 0 to the missing values, followed by a linear scaling of pixel values to the range $[0, 1]$. After that, we downsample the images to $(256, 256)$ using bilinear interpolation. The task is to predict the next sequences of radar echoes in an hour (12 frames) from the given 20-minute radar echoes (4 frames). The data in 2016 and 2017 are sampled as the training set and those in 2018 are sampled as the test set.

**HKO-7.** The HKO-7 dataset [43] is a collection of radar reflectivity image data from 2009 to 2015 based on the radar product, namely, the Constant Altitude Plan Position Indicator (CAPPI) at an altitude of 2 kilometers with a radius of 256 kilometers centered at Hong Kong. No prior data cleansing was applied to the HKO-7 dataset so it may consist of noises commonly found in radar imagery due to sea or ground clutters and anomalous propagation, and blind sectors due to blockage of microwave signals. Moreover, the sub-tropical climate of Hong Kong, mesoscale weather development which is caused by the land-sea contrast and complex terrain over the territory and the adjacent coastal areas lead to changeable weather and limited predictability of severe convective precipitation beyond the next couple of hours. Overall, the HKO-7 dataset is known to be much more difficult to model precisely, which could better highlight the effectiveness of our methods. We predict the next 2-hour radar echoes (20 frames) from that of the past 30 minutes (5 frames). The data from 2009 to 2014 are used as the training set and those in 2015 are used as the test set.

# B   Triviality of $L_2$ Loss in the Fourier Space

In this section, we prove that the $L_2$ distance between ground truth and prediction in both the Fourier domain and image domain are equivalent from both the forward and gradient aspects.

## B.1   Showing that $L_2(\hat{F}, F) = L_2(\hat{X}, X)$

Parseval's Theorem (or the general one: Plancherel Theorem) describes the unitarity of the Fourier transform under proper normalization. Without normalization, we have the following relationship:

$$\sum_{k=0}^{N-1} |X_k|^2 = \frac{1}{N} \sum_{k=0}^{N-1} |F_k|^2.$$

This refers to the 1D case where $F$ is the Fourier transformed output of $X$, and $N$ is the vector length of both $F$ and $X$. In the 2D case with $F$ orthonormalized in Eq. (2), we have instead

$$\sum |X|^2 = \sum |F|^2,$$

where $\sum$ is used as a shorthand summing every element in the following 2D matrix. When we apply the $L_2$ loss to two orthonormalized Fourier matrices, we obtain

$$L_2(F, \hat{F}) = \frac{1}{N} \sum |F - \hat{F}|^2 = \frac{1}{N} \sum (X - \hat{X})^2 = L_2(X, \hat{X}) \tag{9}$$

due to the linearity of the Fourier transform and the use of Parseval's Theorem.

## B.2   Showing that $\frac{\partial L_2(F,\hat{F})}{\partial \hat{X}_{kl}} = \frac{\partial L_2(X,\hat{X})}{\partial \hat{X}_{kl}}$

From Eq. (2), we continue and derive the gradient of the Fourier transform output $F$ with respect to image $X$:

$$\frac{\partial F_{pq}}{\partial X_{kl}} = \frac{1}{\sqrt{MN}} e^{-i2\pi(\frac{kp}{M} + \frac{lq}{N})}$$

For every complex vector, the multiplication of itself and its conjugate is equal to the square of its amplitude, that is $FF^* = |F|^2$. Thus,

$$\frac{\partial}{\partial \hat{X}_{kl}} |F_{pq} - F_{pq}^*|^2 = \frac{\partial}{\partial \hat{X}_{kl}} [|F_{pq}|^2 - F_{pq}^* \hat{F}_{pq} - F_{pq} \hat{F}_{pq}^* + \hat{F}_{pq} \hat{F}_{pq}^*]$$

$$= \frac{1}{\sqrt{MN}} [-F_{pq}^* e^{-i2\pi(\frac{kp}{M} + \frac{lq}{N})} - F_{pq} e^{i2\pi(\frac{kp}{M} + \frac{lq}{N})} + \hat{F}_{pq}^* e^{-i2\pi(\frac{kp}{M} + \frac{lq}{N})} + \hat{F}_{pq} e^{i2\pi(\frac{kp}{M} + \frac{lq}{N})}].$$

With the inverse Fourier transform defined in Eq. (2) and the assumption that $X$ is always real, we can obtain:

$$\frac{\partial L_2(F, \hat{F})}{\partial \hat{X}_{kl}} = \frac{1}{MN} \sum_{p=0}^{M-1} \sum_{q=0}^{N-1} \frac{\partial}{\partial \hat{X}_{kl}} |F_{pq} - F_{pq}^*|^2$$

$$= -\frac{1}{(\sqrt{MN})^3} \sum_{p=0}^{M-1} \sum_{q=0}^{N-1} [F_{pq}^* e^{-i2\pi(\frac{kp}{M} + \frac{lq}{N})} + F_{pq} e^{i2\pi(\frac{kp}{M} + \frac{lq}{N})}$$

$$- \hat{F}_{pq}^* e^{-i2\pi(\frac{kp}{M} + \frac{lq}{N})} - \hat{F}_{pq} e^{i2\pi(\frac{kp}{M} + \frac{lq}{N})}]$$

$$= -\frac{1}{MN} [X_{kl}^* + X_{kl} - \hat{X}_{kl}^* - \hat{X}_{kl}]$$

$$= -\frac{2}{MN} [X_{kl} - \hat{X}_{kl}]$$

$$\frac{\partial L_2(F, \hat{F})}{\partial \hat{X}_{kl}} = \frac{\partial L_2(X, \hat{X})}{\partial \hat{X}_{kl}}$$

Consequently, the gradients of $L_2(F, \hat{F})$ and $L_2(X, \hat{X})$ with respect to $\hat{X}_{kl}$ are equivalent. This result indicates that implementing $L_2$ in the Fourier domain without any weighting as a loss function does not affect the model performance.

## C  FAL in the Gradient Aspect

In Section 3.2, we claim that FAL works as a regularizer to maintain the frequency amplitude in the Fourier domain, but not the full loss function. In this section, we study the reason behind this statement from the perspective of gradient feedback. Before that, we start with deriving the derivative of $|F|$ with respect to $X_{kl}$:

$$|F_{pq}|^2 = F_{pq}F_{pq}^*$$

$$2|F_{pq}|\frac{\partial|F_{pq}|}{\partial X_{kl}} = F_{pq}\frac{\partial F_{pq}^*}{\partial X_{kl}} + \frac{\partial F_{pq}}{\partial X_{kl}}F_{pq}^*$$

$$\frac{\partial|F_{pq}|}{\partial X_{kl}} = \frac{1}{2\sqrt{MN}}[e^{i(\theta_{pq}+\alpha_{pq,kl})} + e^{-i(\theta_{pq}+\alpha_{pq,kl})}]$$

$$= \frac{1}{\sqrt{MN}}\cos(\theta_{pq} + \alpha_{pq,kl}),$$

where $F_{pq} = |F_{pq}|e^{i\theta_{pq}}$ and $\alpha_{pq,kl} = 2\pi(\frac{kp}{M} + \frac{lq}{N})$.

From Eq. (4), we further derive its derivative with respect to $\hat{X}_{kl}$ and get:

$$\frac{\partial}{\partial\hat{X}_{kl}}\text{FAL}(X,\hat{X}) = -\frac{2}{(\sqrt{MN})^3}\sum_{p=0}^{M-1}\sum_{q=0}^{N-1}(|F_{pq}| - |\hat{F}_{pq}|)\cos(\hat{\theta}_{pq} + \alpha_{pq,kl}), \tag{10}$$

where $|F_{pq}|$ and $|\hat{F}_{pq}|$ are the Fourier amplitudes of the ground truth and prediction corresponding to the frequency $(p,q)$ while $\hat{\theta}_{pq}$ is the Fourier phase of prediction corresponding to the frequency $(p,q)$.

From Eq. (10), we note that $\theta_{pq}$, which corresponds to the position or the image structure of the ground truth frequencies (object) in the image space, is missing. In other words, the model never gets its parameters updated based on the phase of the ground truth. As a result, FAL only encourages the model to predict what has the same amplitude distribution in the Fourier domain, without considering the image structure. This theoretically shows the infeasibility of reconstructing the ground truth based on FAL alone, motivating us to adopt a second loss term to maintain the image structure.

To effectively make use of FAL as a regularizer, we have to ensure that the model has sufficient time to learn the general image structure (the low-frequency pattern) such that FAL could provide better guidance on the remaining frequency components by exposing it more to FCL in the beginning of training process. In contrast, if the model cannot learn the low-frequency components before FAL becomes the dominant learning objective, the model will likely converge to a trivial solution. This claim is also empirically verified by our ablation study experiments where using FAL alone results in poor performance as shown in Appendix. F.

# D  Further Analysis of FAL

This section discerns FAL and a naive $L_2$ loss in the Fourier space. By definition, the major difference between the two is whether the complex pattern or the amplitude is used in the comparison. The FAL loss term can be expanded as follows:

$$
\begin{aligned}
\text{FAL}(X, \hat{X}) &= \frac{1}{MN} \sum_{p=0}^{M-1} \sum_{q=0}^{N-1} (|F|_{pq} - |\hat{F}|_{pq})^2 \\
&= \frac{1}{MN} \sum_{p=0}^{M-1} \sum_{q=0}^{N-1} (|F|_{pq}^2 + |F|_{pq}^2 - 2|F|_{pq}|\hat{F}|_{pq}) \\
&= \frac{1}{MN} \sum_{p=0}^{M-1} \sum_{q=0}^{N-1} (X_{pq}^2 + \hat{X}_{pq}^2) - \frac{1}{MN} \sum_{p=0}^{M-1} \sum_{q=0}^{N-1} 2|F|_{pq}|\hat{F}|_{pq} \\
&= \frac{1}{MN} \sum_{p=0}^{M-1} \sum_{q=0}^{N-1} (X_{pq}^2 + \hat{X}_{pq}^2 - 2X_{pq}\hat{X}_{pq}) + \\
&\quad \frac{1}{MN} \sum_{p=0}^{M-1} \sum_{q=0}^{N-1} 2\hat{X}_{pq}X_{pq} - \frac{1}{MN} \sum_{p=0}^{M-1} \sum_{q=0}^{N-1} 2|F|_{pq}|\hat{F}|_{pq} \\
&= L_2(X, \hat{X}) + \frac{1}{MN} \sum_{p=0}^{M-1} \sum_{q=0}^{N-1} 2X_{pq}\hat{X}_{pq} - \frac{1}{MN} \sum_{p=0}^{M-1} \sum_{q=0}^{N-1} 2|F|_{pq}|\hat{F}|_{pq}.
\end{aligned}
$$

Apart from the $L_2$ component which is equivalent to Eq. (9), we also obtain two extra terms, shorthanded as $\sum 2X\hat{X}$ and $\sum 2|F||\hat{F}|$. To study the empirical effect of the two factors on the high-frequency components, we performed a simple experiment: we sampled an image from the Moving-MNIST dataset and performed two modifications over time: (1) applying Gaussian blur with a standard deviation of $\sigma$ to the sample, and (2) translating the sample along the direction $(t, t)$. Then we observed the trend of increment of the two factors, as shown in Figure 4.

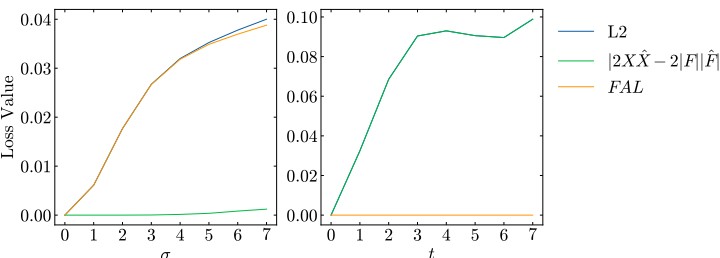

Figure 4: FAL loss terms over different values of (left) $\sigma$ in blurring and (right) $t$ in translation. In (right), $L_2$ (the blue line) and $|\sum 2X\hat{X} - \sum 2|F||\hat{F}||$ (the green line) mostly overlap.

The figure reflects a couple of characteristics of the FAL loss term. First, in the case of blurring, it behaves similarly to the standard $L_2$ loss with a tiny difference when $\sigma$ gets very large. Intriguingly, FAL does not exhibit a different degree of sensitivity to different frequencies. However, for translation, the absolute difference $\sum 2X\hat{X} - \sum 2|F||\hat{F}|$ is almost equivalent to and thus cancels out the $L_2$ loss, causing the final FAL loss term to become very small. It is also noteworthy that when the two samples $X$ and $\hat{X}$ are identical, both $\sum 2X\hat{X}$ and $\sum 2|F||\hat{F}|$ are zero and thus FAL is also zero. From the observations above, FAL is invariant to global translations and robust to one-directional local translation compared to $L_2$. Because of such invariance, FAL is more robust against the spectral bias and could better fine-tune the frequencies of the output. Utilizing this behavior, the model could focus on the reconstruction of a clear signal without suffering from the influence of the randomness in translations. Moreover, it also shows that an arbitrary scaling between $L_2$ and the factor $\sum 2X\hat{X} - \sum 2|F||\hat{F}|$ could not result in a desired effect, since this causes the two terms to no longer overlap in the plot over $t$, leading to an increase in sensitivity to translation.

# E    Further Analysis of FCL

To understand how FCL affects the model during training, we derive its derivative with respect to $\hat{X}_{kl}$ and obtain an interesting result with the aid of Plancherel's theorem:

$$\frac{\partial}{\partial \hat{X}_{kl}} \text{FCL}(X, \hat{X}) = -\frac{1}{\sqrt{\sum |X|^2 \sum |\hat{X}|^2}} [X_{kl} - \frac{\sum X\hat{X}}{\sum |\hat{X}|^2} \hat{X}_{kl}] \qquad (11)$$

From Eq. (11), the ratio $X_{kl}$ to $\hat{X}_{kl}$ highly depends on the summation over the image domain, providing global information to $\hat{X}_{kl}$, unlike the element-wise or pixel-wise relationship between $\hat{X}$ and $X$ in the conventional MSE loss, $L_2(\hat{X}, X)$.

To have an intuitive understanding of the conclusion above, we design a thought experiment to understand how FCL is different from $L_2(\hat{X}, X)$ here. Consider the case where the prediction has the same image structure as the ground truth but with different intensity, for instance, $\hat{X} = \beta X$, where $\beta$ is an arbitrary non-zero constant.

Substituting $\hat{X}$ into Eq. (11), we have

$$\frac{\partial}{\partial \hat{X}_{kl}} \text{FCL}(X, \hat{X}) = 0.$$

Meanwhile, it is straightforward that

$$\frac{\partial}{\partial \hat{X}_{kl}} L_2(\hat{X}, X) \propto -(1 - \beta).$$

With the above discrepancy, the behavior of FCL is substantially different from MSE in regard to overall brightness. That is, MSE is affected by both the image structure and the overall brightness but FCL is affected by the image structure only. Therefore, with FCL alone, we lose the pixel intensity. While applying the sigmoid function is one method to alleviate the drawback of the missing information, incorporating FAL which focuses on the intensity in particular could be viewed as a parallel complement to further stabilize the models.

# F    Ablation Study on FAL and FCL

In previous sections, we showed that the Fourier phase of the ground truth, $\theta$, is missing in the gradient of FAL. Hence, we claim that FAL alone is insufficient to be a reconstruction loss. Similarly, in the thought experiment conducted in Appendix E, we conclude that FCL does not consider the image intensity and sharpness. As a result, the two loss terms FAL and FCL have to be used together as a full reconstruction loss. Here, we report the empirical results of the ablation study for FAL and FCL in Table 3.

Table 3: Quantitative performance of different losses for ConvLSTM on Stochastic Moving-MNIST.

| Loss | Metrics | | | | | | |
|---|---|---|---|---|---|---|---|
| | MAE↓ | MSE↓ | SSIM↑ | LPIPS↓ | FVD↓ | FSS↑ | RHD↓ |
| FAL Only | 430.7 | 302.7 | 0.2871 | 0.5854 | 1320.1 | 0.0019 | 1.0538 |
| FCL Only | 184.9 | **80.4** | 0.7318 | 0.2102 | 391.8 | 0.6451 | 1.0841 |
| FACL | **180.1** | 118.1 | **0.7463** | **0.1092** | **82.3** | **0.8172** | **0.3391** |

In Table 3, the model trained with FAL does not produce meaningful output, as reflected by the abnormal values of the metrics and the faulty predictions in Figure 5. This agrees with our statement that models trained with FAL alone cannot converge to proper local minima. Meanwhile, the model trained with FCL only exhibits behaviors similar to MSE as shown in Figure 5. To sum up, either using FAL or FCL alone does not empirically produce the desired effect. However, combining the two loss terms together achieves a huge improvement to most of the metrics, which agrees with our theoretical analysis.

Next, we study the effect of $\alpha$, which controls the length of the fine-tuning process with FAL. The fine-tuning process encourages the models to predict sharper and brighter predictions. At the same

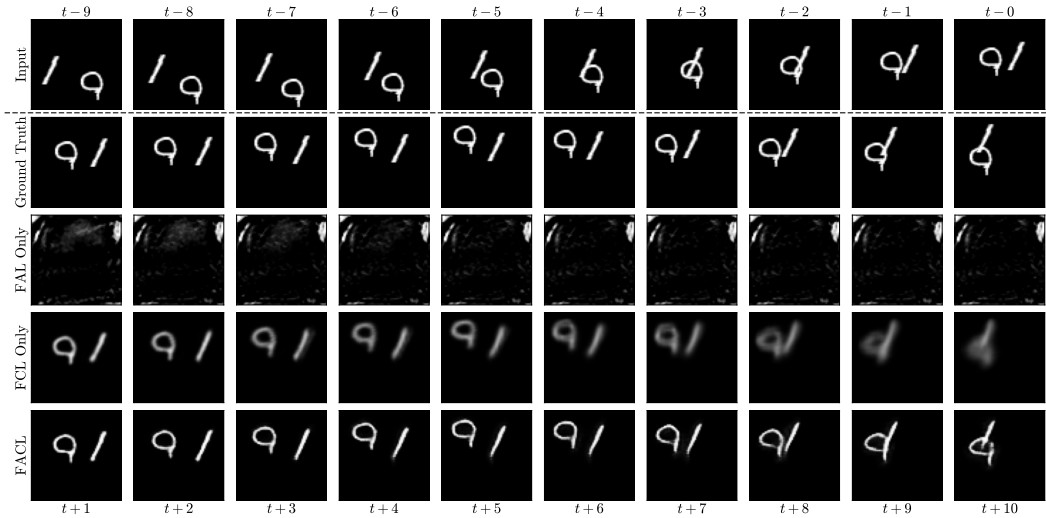

Figure 5: Qualitative performance of different losses for ConvLSTM on Stochastic Moving-MNIST.

time, this can also lead to overfitting of noises in the high-frequency components. From Table 4 and 5, the sharpness-aware metrics such as LPIPS, FVD and RHD can always be improved by setting $\alpha$ to non-zero. However, there is no apparent improvement or decay after $\alpha \geq 0.2$. Therefore, we believe setting $\alpha = 0.1$ or $0.2$ as a default value can strike a good balance between sharpness and pixel-wise performance.

Table 4: Effect of different $\alpha$ on the performance of PredRNN trained with FACL on Stochastic Moving-MNIST.

| $\alpha$ | Metrics | | | | | | |
|---|---|---|---|---|---|---|---|
| | MAE↓ | MSE↓ | SSIM↑ | LPIPS↓ | FVD↓ | FSS↑ | RHD↓ |
| 0.0 | 162.7 | **93.76** | 0.7770 | 0.0997 | 92.53 | 0.8533 | 0.3584 |
| 0.1 | 162.8 | 104.3 | 0.7759 | 0.0817 | 62.16 | 0.8534 | 0.2982 |
| 0.2 | **162.0** | 105.0 | **0.7822** | **0.0806** | **54.63** | **0.8552** | **0.2976** |
| 0.3 | 162.1 | 105.2 | 0.7812 | 0.0869 | 63.29 | 0.8549 | 0.3000 |
| 0.4 | 162.6 | 105.3 | 0.7806 | 0.0845 | 58.95 | 0.8542 | 0.3023 |

Table 5: Effect of different $\alpha$ on the performance of ConvLSTM trained with FACL on SEVIR, where MAE is in the scale of $10^{-3}$.

| $\alpha$ | Metrics | | | | | | | | |
|---|---|---|---|---|---|---|---|---|---|
| | MAE↓ | SSIM↑ | LPIPS↓ | FVD↓ | CSI-m↑ | CSI$_4$-m↑ | CSI$_{16}$-m↑ | FSS↑ | RHD↓ |
| 0.0 | **26.15** | **0.7814** | 0.3502 | 391.37 | **0.4195** | **0.4339** | 0.4710 | **0.5727** | 1.3924 |
| 0.1 | 27.60 | 0.7624 | 0.3508 | 289.49 | 0.3984 | 0.4295 | 0.5073 | 0.5640 | 1.2087 |
| 0.2 | 27.92 | 0.7589 | 0.3415 | 254.73 | 0.3958 | 0.4331 | **0.5247** | 0.5447 | **1.1643** |
| 0.3 | 27.80 | 0.7587 | **0.3312** | 258.24 | 0.3953 | 0.4288 | 0.5242 | 0.5453 | 1.1710 |
| 0.4 | 28.15 | 0.7574 | 0.3384 | **232.50** | 0.3930 | 0.4264 | 0.5190 | 0.5262 | 1.1960 |
| 0.5 | 30.45 | 0.7402 | 0.3492 | 281.82 | 0.3633 | 0.3915 | 0.4838 | 0.4813 | 1.3445 |

# G Running Time of FACL

In the previous sections, we showed that the method is both effective and generic of models. In this section, we discuss the running time of FACL. Theoretically, FACL utilizes DFT, which has time complexity $O(n^2)$ in the 1D case with vector length $n$. By leveraging the 2D Fast Fourier Transform (FFT), we could improve the time complexity to $O(MN(\log(M) + \log(N)))$ for each pair of frames, where $M$ and $N$ correspond to the height and width of the samples. With the aid of deep learning frameworks such as PyTorch, such operations can be run in parallel and supported by GPU. Therefore, the computational load for FACL is light compared to the deep network architectures. During inference, the only difference between models trained with MSE and FACL is that the FACL one consists of a sigmoid layer at the end. Running in parallel, again, this operation is negligible.

To test the actual speed of FACL, we report the running time during model training and model inference for the experimented models in Table 6. For the training stage, we report the mean of the training time for the first 5 epochs. The table shows that the running time of FACL is negligible compared to the MSE counterpart, with the model selected being the most dominant factor for the running time. With the inference time reported, we could also notice the advantage of staying with video prediction models over generative models. The diffusion models are much slower than traditional predictive models. Our slowest model (FACL on PredRNN) is still 50X faster than MCVD. Note that such a difference scales with the image size, causing some of the generative models infeasible to apply to large-size radar imagery.

Table 6: Comparison of the quantitative performance of different losses for models trained on Stochastic Moving-MNIST datasets. We report the average time (in seconds) of 5 training epochs and 100 inference steps on a single Nvidia GeForce RTX3090.

| Model | Loss | Training Time (s) | Inference Time (s) | Average FPS |
|---|---|---|---|---|
| ConvLSTM | MSE | 97.8 | 0.045 | 220 |
| ConvLSTM | FACL | 97.8 | 0.043 | 232 |
| PredRNN | MSE | 132.6 | 0.169 | 59 |
| PredRNN | FACL | 134.2 | 0.180 | 55 |
| SimVP | MSE | 36.4 | 0.022 | 635 |
| SimVP | FACL | 29.8 | 0.017 | 598 |
| Earthformer | MSE | 724.5 | 0.101 | 99 |
| Earthformer | FACL | 731.3 | 0.110 | 91 |
| LDCast | - | - | 7.783 | 1.3 |
| MCVD | - | - | 81.873 | 0.12 |

# H Evaluation on N-Body-MNIST

Apart from the proposed Stochastic Moving-MNIST, previous works proposed N-Body-MNIST [10], a dataset that utilizes multiple transformations to simulate the chaotic nature of the atmospheric conditions. We present the results in this section.

Table 7: Comparison of the quantitative performance of different losses for models trained on the Stochastic Moving-MNIST. The better score between MSE and FACL is highlighted in bold.

| Model | Loss | Pixel-wise/Structural | | Perceptual | | Skill | Proposed |
|---|---|---|---|---|---|---|---|
| | | MAE↓ | SSIM↑ | LPIPS↓ | FVD↓ | FSS↑ | RHD↓ |
| ConvLSTM | MSE | 57.2 | 0.8946 | 0.1264 | 178.57 | 0.7601 | 0.2301 |
| | FACL | **43.1** | **0.9385** | **0.0533** | **80.83** | **0.9198** | **0.1586** |
| SimVP | MSE | **55.2** | **0.9130** | **0.0612** | **77.95** | **0.9093** | **0.1467** |
| | FACL | 59.3 | 0.8960 | 0.0730 | 102.71 | 0.8978 | 0.1526 |
| Earthformer | MSE | **18.6** | **0.9834** | **0.0091** | **13.46** | **0.9835** | **0.0971** |
| | FACL | 19.3 | 0.9826 | 0.0092 | 13.68 | 0.9829 | 0.0985 |

Compared with Stochastic Moving-MNIST, N-Body-MNIST additionally introduces inter-digit influence on top of the original trajectory. However, from the table and visualizations, we observe that the models in general can perform much better with N-Body-MNIST than that with Stochastic Moving-MNIST. This shows our assumption that traditional video prediction models can capture

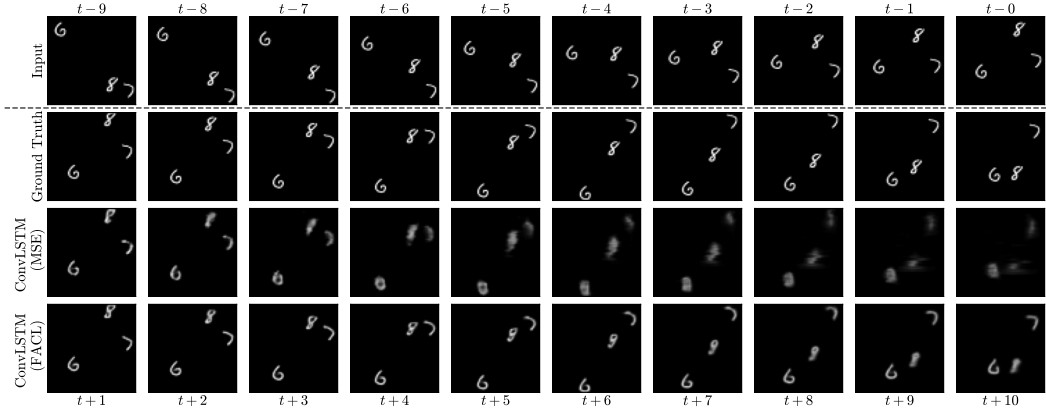

Figure 6: Output frames of ConvLSTM trained with MSE and FACL on N-body-MNIST.

complicated deterministic motion but cannot handle random motion well. Hence, despite having N-body-MNIST as a benchmark dataset, the proposal of Stochastic Moving-MNIST is still necessary to study the models' characteristics in handling random motion. With a highly deterministic dataset consisting of tiny digits, strong models like Earthformer and SimVP can almost perfectly grasp the motion and thus result in an excellent performance. Under such scenario, switching to FACL does not bring further improvement to the models.

## I  Evaluation Results for PredRNN

This section is an extension of Table 2 featuring PredRNN. Due to the limitation of the model and consideration of training efficiency, we downscale the radar data to $128 \times 128$ with bilinear interpolation. The results are reported in Table.8. Note that the image resolution influences some metrics such as LPIPS and CSI with pooling, causing an unfair comparison with the results in Table 2.

Table 8: Comparison of the quantitative performance of different losses for PredRNN trained on SEVIR, MeteoNet and HKO-7. MAE is in the scale of $10^{-3}$. The better score between MSE and FACL is highlighted in bold.

| Dataset | Loss | Pixelwise/Structural | | Perceptral | | Skill | | | | Proposed |
| | | MAE↓ | SSIM↑ | LPIPS↓ | FVD↓ | CSI-m↑ | CSI$_4$-m↑ | CSI$_{16}$-m↑ | FSS↑ | RHD↓ |
|---|---|---|---|---|---|---|---|---|---|---|
| SEVIR | MSE | **28.91** | **0.7238** | 0.3572 | 528.8 | 0.3553 | 0.3702 | 0.4153 | 0.5552 | 1.1333 |
| | FACL | 31.37 | 0.7083 | **0.3206** | **384.2** | **0.3553** | **0.4176** | **0.5378** | **0.5830** | **0.8492** |
| MeteoNet | MSE | **7.39** | **0.9016** | 0.1675 | 375.7 | **0.4348** | 0.4472 | 0.4981 | **0.5835** | 0.2207 |
| | FACL | 8.37 | 0.8935 | **0.1346** | **214.2** | 0.3690 | **0.4873** | **0.6313** | 0.5570 | **0.1515** |
| HKO-7 | MSE | **23.79** | 0.7081 | 0.3003 | 503.2 | 0.3168 | 0.3070 | 0.3213 | **0.4982** | 0.7571 |
| | FACL | 24.38 | **0.7174** | **0.2617** | **359.6** | **0.3398** | **0.3908** | **0.4870** | 0.4797 | **0.5339** |

## J  Comparison with Other Loss Alternatives

Apart from the previous works discussed in Section 2, there has also been a series of attempts to improve the loss function. For instance, Balanced Mean Squared Error (BMSE) [43] was proposed to increase the weighting on heavy rainfall. Multi-sigmoid loss (SSL) [48] preprocesses the images with linear transformations and nonlinear sigmoid function before applying MSE. Tran et al. [49] tested SSIM and MS-SSIM and recommended MSE+SSIM to be the loss function. We also present the results by adopting these losses with ConvLSTM trained on Stochastic Moving MNIST. For SSL, we follow the paper by picking $i \in \left[\frac{20}{70}, \frac{30}{70}\right]$ and $c = 20$.

One point worth noting is that none of the methods other than FACL generates a sharp prediction qualitatively, despite occasionally higher pixel-wise performance. Besides, we can draw the following conclusions:

- SSL improves the model performance in general, but still cannot generate clear output under stochastic motion.

Table 9: Comparison of the quantitative performance of different losses for ConvLSTM trained on Stochastic Moving-MNIST.

| Loss | Metrics | | | | | |
|------|------|------|------|------|------|------|
| | MAE | SSIM | LPIPS | FVD | FSS | RHD |
| MSE | 196.42 | 0.6975 | 0.2538 | 451.54 | 0.6148 | 1.1504 |
| SSL [48] | **175.17** | **0.7553** | 0.1906 | 348.18 | 0.7225 | 0.9840 |
| MSE+SSIM [49] | 184.10 | 0.7488 | 0.2573 | 529.71 | 0.3514 | 0.7921 |
| FACL | 180.10 | 0.7463 | **0.1092** | **82.28** | **0.8172** | **0.3391** |

Table 10: Comparison of the quantitative performance of different losses for ConvLSTM trained on HKO-7.

| Loss | Metrics | | | | | | | | |
|------|------|------|------|------|------|------|------|------|------|
| | MAE | SSIM | LPIPS | FVD | CSI-m | CSI4-m | CSI16-m | FSS | RHD |
| MSE | 30.43 | 0.6664 | 0.3057 | 791.3 | 0.2772 | 0.2282 | 0.1702 | 0.2653 | 1.2453 |
| BMSE [43] | 45.03 | 0.5537 | 0.3804 | 901.9 | **0.3484** | **0.3670** | **0.3354** | 0.3999 | 1.7918 |
| FACL | **29.72** | **0.7168** | **0.2962** | **569.1** | 0.3054 | 0.3040 | 0.3351 | **0.7916** | **0.4045** |

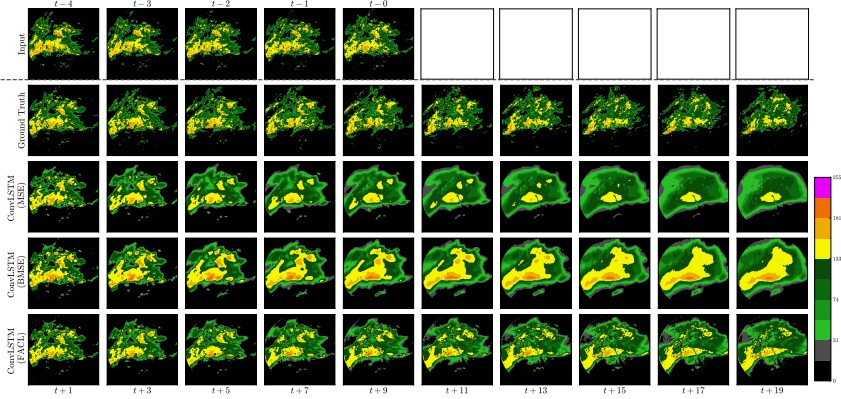

Figure 7: Output frames of ConvLSTM trained with MSE, SSL MSE+SSIM and FACL on Stochastic Moving-MNIST.

Figure 8: Output frames of ConvLSTM trained with MSE, BMSE and FACL on HKO-7.

- Losses integrating SSIM (and also L1) "dissolves" the prediction to zero over time under uncertainty. Such an effect is especially significant for weaker models.
- Weighted MSE (such as BMSE) only "tilts" the focus. BMSE severely over-predicts in exchange for an improvement in CSI, sacrificing all other metrics such as MAE, SSIM, LPIPS, FVD, FSS and RHD.

# K   Applying FACL to Generative Models

Apart than replacing the MSE loss in video prediction, we also study the scenario where FACL is applied to video generative models. Specifically, we have tested three generative models using different generative methods: MCVD [21], a diffusion-based model; STRPM [45], a GAN-based model training a recurrent generator; and SVGLP [46], a VAE-based model with its default loss function being a composition of the MSE term and KL-divergence term. We replace the MSE term in each of these models with FACL, and report the quantitative performance in Table 11, while its visualization is reported in Figure 9.

In the table, FACL exhibits to be a good substitute for MSE even in the generative models as it improves most of the metrics. For both SVGLP and STRPM, replacing the reconstruction loss with FACL further improves the image quality of the prediction, as reflected by the significant drop in FVD and RHD and vast improvement in FSS. On the other hand, using FACL in MCVD is much less intuitive as the MSE loss fits the diffusion output to Gaussian noise. Since there is no point in studying the noise frequencies, the performance gain attributed to FACL appears trivial, resulting in comparable performance between MSE and FACL.

Table 11: Quantitative performance of SVGLP, STRPM and MCVD with different loss, trained on the Stochastic Moving-MNIST.

| Model | Loss | Metrics | | | | | |
|-------|------|------|------|------|------|------|------|
| | | MAE↓ | SSIM↑ | LPIPS↓ | FVD↓ | FSS↑ | RHD↓ |
| SVGLP | MSE + $D_{\text{KL}}$ | 209.5 | 0.7300 | 0.1412 | 136.80 | 0.7156 | 0.6031 |
| | FACL + $D_{\text{KL}}$ | **201.1** | **0.7377** | **0.1080** | **62.70** | **0.7458** | **0.3871** |
| STRPM | MSE + $L_{\text{LP}}$ + $L_{\text{GAN}}$ | **154.0** | **0.7912** | 0.1017 | 117.35 | 0.8337 | 0.3216 |
| | FACL + $L_{\text{LP}}$ + $L_{\text{GAN}}$ | 161.9 | 0.7849 | **0.0960** | **91.97** | **0.8453** | **0.3113** |
| MCVD | $L_{\text{vidpred}}$ | 219.9 | **0.7125** | **0.1033** | 44.70 | 0.7184 | 0.3941 |
| | FACL | **219.6** | 0.7051 | 0.1041 | **42.40** | **0.7251** | **0.3897** |

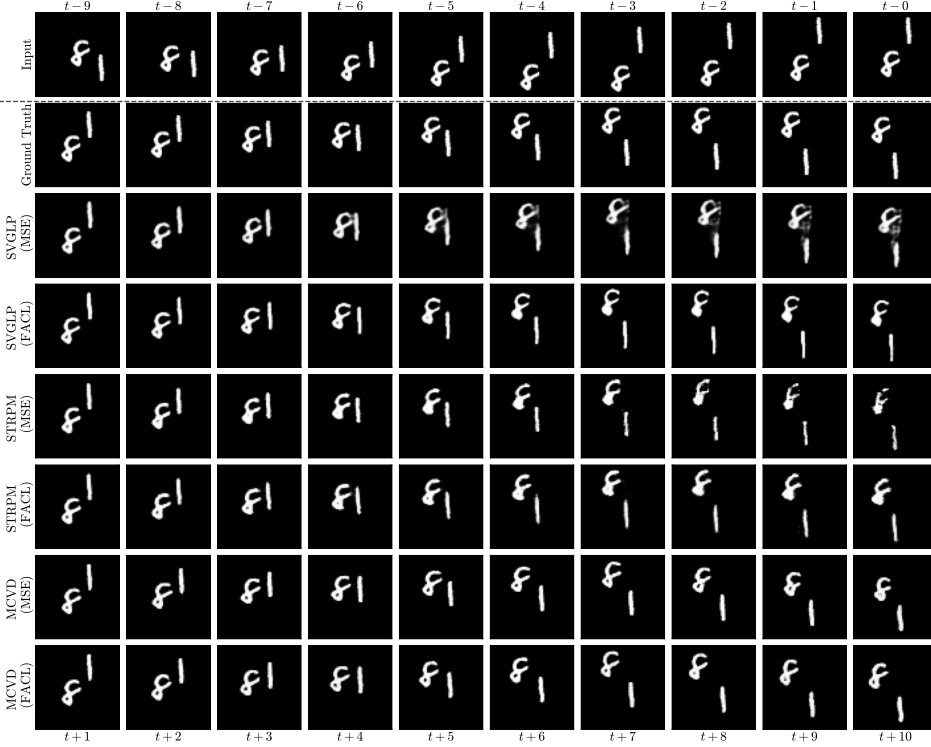

Figure 9: Output frames of video generative models trained with different losses stated in Table 11 on Stochastic Moving-MNIST.

## L  Analysis and Discussion of RHD

In the paper, we utilized three types of metrics to measure the similarity between two image sequences: pixel-wise and structural metrics, perceptual metrics and meteorological skill scores. Each suffers from its kind of drawbacks. For example, pixel-wise differences such as MAE and MSE do not consider the overall image structure, causing great encouragement to blurry prediction. Perceptual metrics such as LPIPS and FVD are based on pre-trained deep learning models (usually on ImageNet), which can suffer from domain bias that does not favor signal-based images. Moreover, deep perceptual metrics can be insensitive to transformations such as global rotation and spatial flipping.

To study the behavior of these metrics and compare them with RHD, we sample a random precipitation event (as visualized in Figure. 10) and apply the following transformations to the image:

- Gaussian blur with kernel size 27 and $\sigma = 15$.
- Translation by $(4, 4)$.
- Clockwise rotation by $5°$.
- Brightening by 2X if the pixel value is higher than $0.5$.
- Darkening by 2X if the pixel value is lower than $0.5$.

The former three transformations study the robustness of the metric under blurring and transformation. The brightening action simulates forecasts that overestimate and the darkening action simulates forecasts that underestimate. After obtaining the transformed images, we measure the evaluation metrics between the distorted images and their corresponding ground truth. The result is reported in Table. 12. FVD is not computed since it requires a larger set of data to form an image distribution.

In the table, a couple of behaviors deserve to be pointed out. First of all, pixel-wise and structural metrics appear to be insensitive to blur, which exhibits a huge difference compared with the other transformations in Figure 10. Such a characteristic discourages small translation which is undesired for precipitation nowcasting. Perceptual metrics such as LPIPS behave the opposite, where blur is the most penalized and value scaling (brightening and darkening) is the most rewarded transformation. Despite this, we believe LPIPS penalizes too little on brightening and darkening as they could result in wrong alerts for extreme weather. For the skill scores, we again observe that CSI with larger pooling tolerates more translation and rotation. In other words, CSI with a large pooling size can be a good metric to penalize blur. However, since CSI discourages false positives, low-range prediction usually wins in CSI which is undesirable for extreme weather forecast. On the other hand, FSS results in unstable behavior for brightening and darkening, due to a single threshold which causes a huge error within a binary cutoff. Among the metrics, we find that RHD is more robust to spatial and pixel-wise transformation while penalizing blur. With the multi-class behavior of RHD, it is also much more stable without bias over overestimation or underestimation.

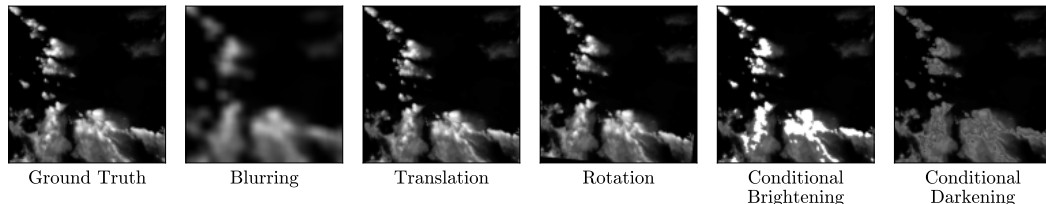

| Ground Truth | Blurring | Translation | Rotation | Conditional Brightening | Conditional Darkening |

Figure 10: Visualization of different transformation techniques applied on the radar image.

Table 12: The values of different metrics on different transformations, where MAE and MSE are in the scale of $10^{-3}$. The worst score for each metric under the tested distortions is underlined and the best score is in bold.

| | Pixel-wise/Structural | | | Perceptual | Skill | | | | Proposed |
|---|---|---|---|---|---|---|---|---|---|
| | MAE↓ | MSE↓ | SSIM↑ | LPIPS↓ | CSI-m↑ | CSI$_4$-m↑ | CSI$_{16}$-m↑ | FSS↑ | RHD↓ |
| Blur | 17.90 | **2.18** | 0.8487 | 0.3660 | 0.5031 | 0.4725 | 0.4210 | 0.5108 | 1.1088 |
| Tran. | 20.40 | 3.73 | 0.8320 | 0.1355 | 0.5582 | 0.6134 | 0.7763 | 0.6782 | 0.6133 |
| Rot. | 32.29 | 8.13 | 0.7767 | 0.2121 | 0.4084 | 0.4520 | 0.6164 | 0.5180 | 1.2650 |
| Brig. | 15.33 | 6.21 | **0.9561** | 0.0778 | 0.5920 | 0.6105 | 0.6675 | **1.0000** | **0.5663** |
| Dark. | **13.08** | 4.26 | 0.9461 | **0.0611** | **0.7597** | **0.7820** | **0.8229** | 0.0781 | 0.5830 |

To sum up, RHD can be viewed as a generalization of FSS or CSI with consideration of multiple radii and pooling sizes. KL-divergence is adopted to measure the similarity of class distribution, replacing the binary segregation used commonly in the meteorological skill scores. Unlike SSIM being a normalized score from 0 to 1, only inspecting the magnitude of RHD is not meaningful. Instead, the users need to specify a fixed set of parameters such as window size and bin ranges, such that relative comparisons between two forecasts under the same set of parameters can provide useful evaluation feedback.

# M    Model Details and Hyper-parameters

This section lists the implementation details of the models and the hyper-parameters used in the experiments described in Section 4.

We followed OpenSTL [13] in implementing PredRNN and SimVP. For PredRNN, apart from the zigzag recurrence, we also adopted scheduled sampling and patch reshaping. For SimVP, we chose the Inception module as the translator in SimVP (also known as SimVP v1). To support varying output sequence lengths, the input to the SimVP decoder is zero-padded to support cases where the output length is larger than the input length. The hyper-parameters are reported in Table 13.

Table 13: Hyper-parameters used for training different models on different datasets. Models trained with both MSE and FACL share the same configuration.

| | Hyper-parameters | Stochastic Moving-MNIST | SEVIR | MeteoNet | HKO-7 |
|---|---|---|---|---|---|
| | Input length | 10 | 13 | 4* | 5 |
| | Output length | 10 | 12 | 12 | 20 |
| | Optimizer | | AdamW | | |
| | $\beta_1$ | | 0.9 | | |
| | $\beta_2$ | | 0.999 | | |
| Common | Weight decay | | 0.01 | | |
| | LR Scheduler | | Cosine Annealing | | |
| | Max LR | 1e-3 | 1e-3 | 1e-3 | 1e-3 |
| | Early stop | | False | | |
| | Training steps | 200 epochs | 50 epochs | 20 epochs | 50K steps |
| ConvLSTM | Batch size | 16 | 4 | 4 | 4 |
| | Image size | $64 \times 64$ | $384 \times 384$ | $256 \times 256$ | $480 \times 480$ |
| | Training steps | 200 epochs | 50 epochs | 20 epochs | 50K steps |
| PredRNN | Batch size | 16 | 4 | 4 | 4 |
| | Image size | $64 \times 64$ | $128 \times 128$ | $128 \times 128$ | $128 \times 128$ |
| | Patch size | $4 \times 4$ | $4 \times 4$ | $4 \times 4$ | $4 \times 4$ |
| | Training steps | 1000 epochs | 50 epochs | 20 epochs | 50K steps |
| SimVP | Batch size | 16 | 4 | 4 | 4 |
| | Image size | $64 \times 64$ | $384 \times 384$ | $256 \times 256$ | $480 \times 480$ |
| | Training steps | 200 epochs | 50 epochs | 20 epochs | 50K steps |
| | Batch size | 32 | 32 | 32 | 32 |
| Earthformer | Image size | $64 \times 64$ | $384 \times 384$ | $256 \times 256$ | $480 \times 480$ |
| | Max LR | 1e-3 | 1e-3 | 1e-3 | 1e-3 |
| | LR Scheduler | | Cosine Annealing | | |
| | Warm-up % | | 20% | | |
| | Input length | 8 | 12 | 4 | 4 |
| | Output length | 8 | 12 | 12 | 20 |
| | Image size | $64 \times 64$ | $384 \times 384$ | $256 \times 256$ | $256 \times 256$ |
| | Optimizer | | AdamW | | |
| | $\beta_1$ | | 0.5 | | |
| LDCast | $\beta_2$ | | 0.9 | | |
| | Weight decay | | 0.001 | | |
| | LR Scheduler | | Reduce-on-plateau | | |
| | patience | | 3 epochs | | |
| | Max LR | | 1e-4 | | |
| | Early stop | | True | | |
| | Training Steps | 1000 epochs | 200 epochs | 50 epochs | 150K steps |
| | Batch Size | 64 | 4 | 8 | 16 |
| MCVD | Image size | $64 \times 64$ | $384 \times 384$ | $256 \times 256$ | $128 \times 128$ |
| | LR Scheduler | | Cosine Annealing | | |
| | Warm-up % | | 20% | | |

* In the case of Earthformer, the input length is set to be 12 regardless of the training loss.

# N    More Qualitative Visualization Comparing with FACL

This section extends the visualizations in Figure 2 and Figure 3 by including the remaining models used in the experiments. Figure 11 visualizes an example output of the remaining models on Stochastic Moving-MNIST and Figure 12 visualizes that of SEVIR. In addition, we further plot an event from HKO-7 and MeteoNet, as shown in Figure 13 and Figure 14 respectively.

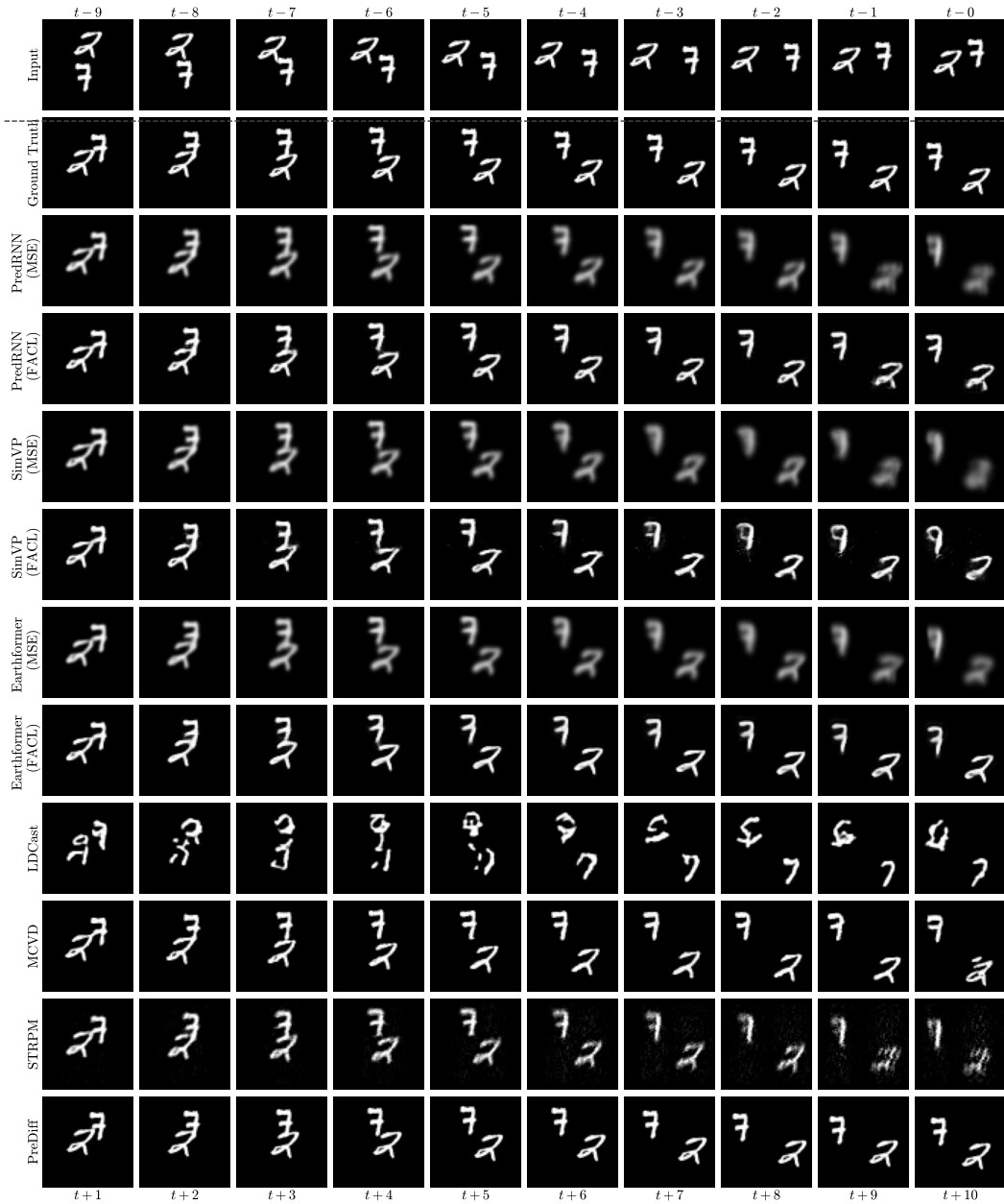

Figure 11: Output frames of the experimented model trained with different losses on Stochastic Moving-MNIST. The extra frames of LDCast are generated with auto-regressive inference.

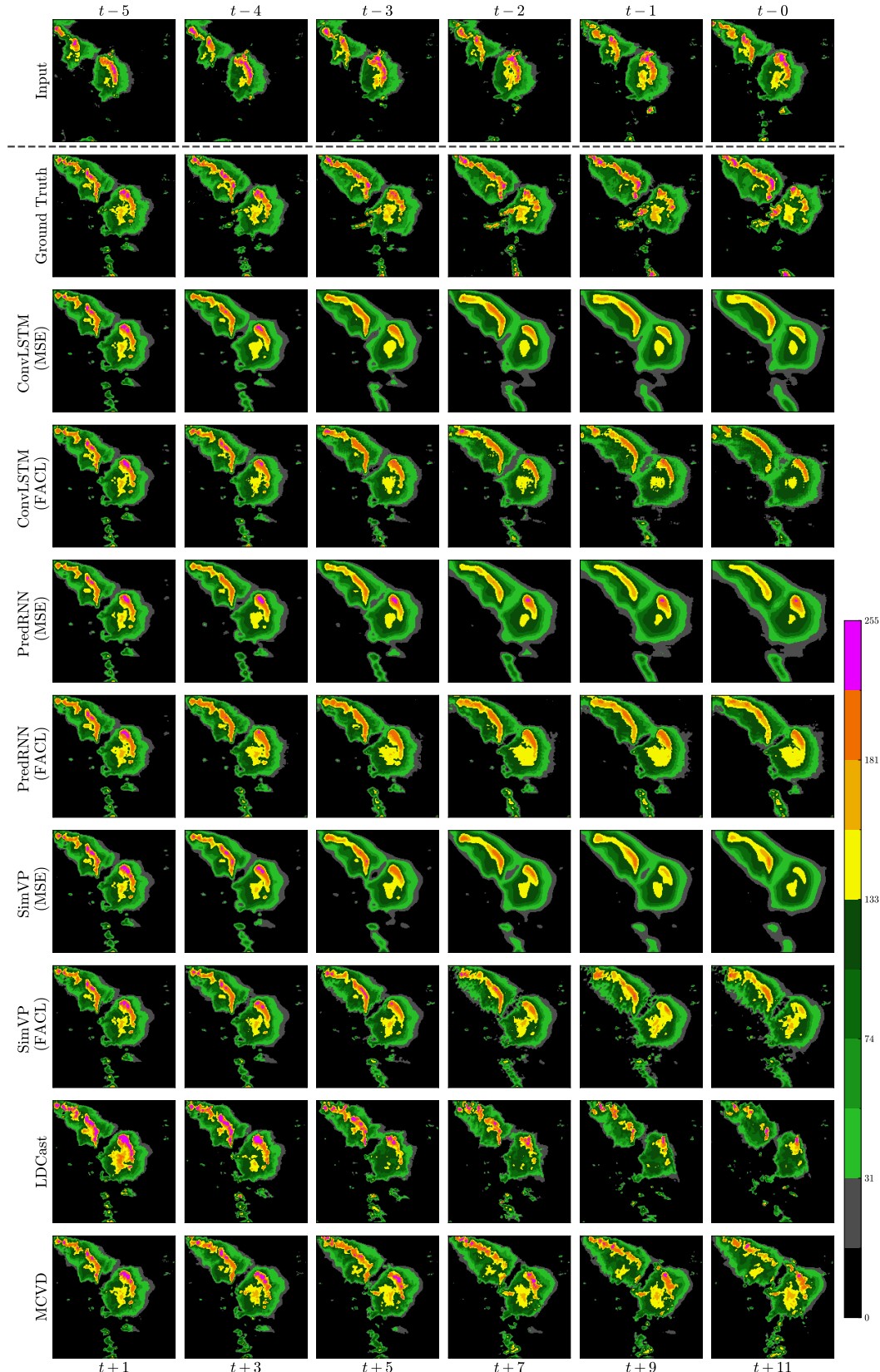

Figure 12: Output frames of the experimented model trained with different losses on SEVIR.

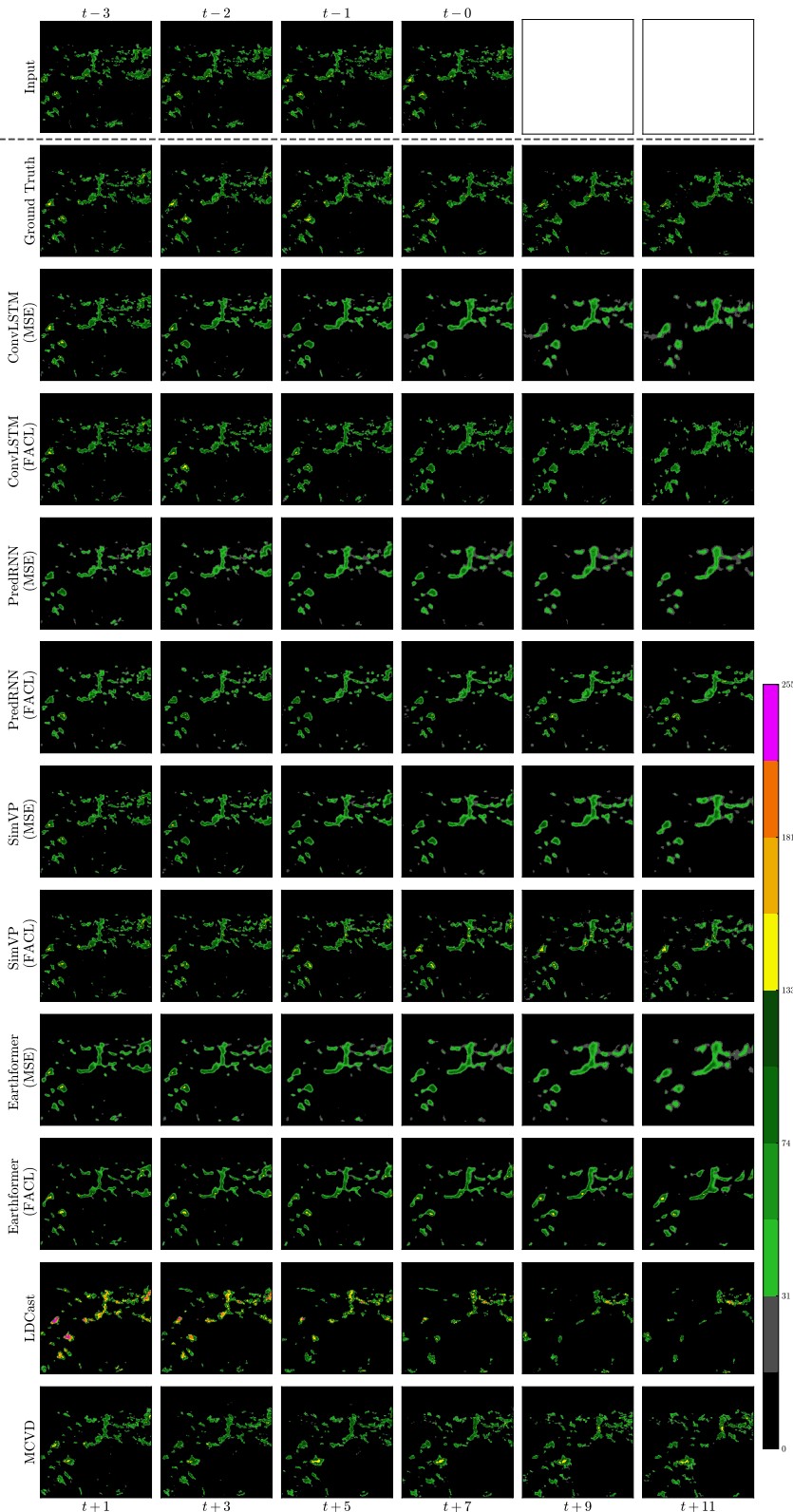

Figure 13: Output frames of the experimented model trained with different losses on MeteoNet.

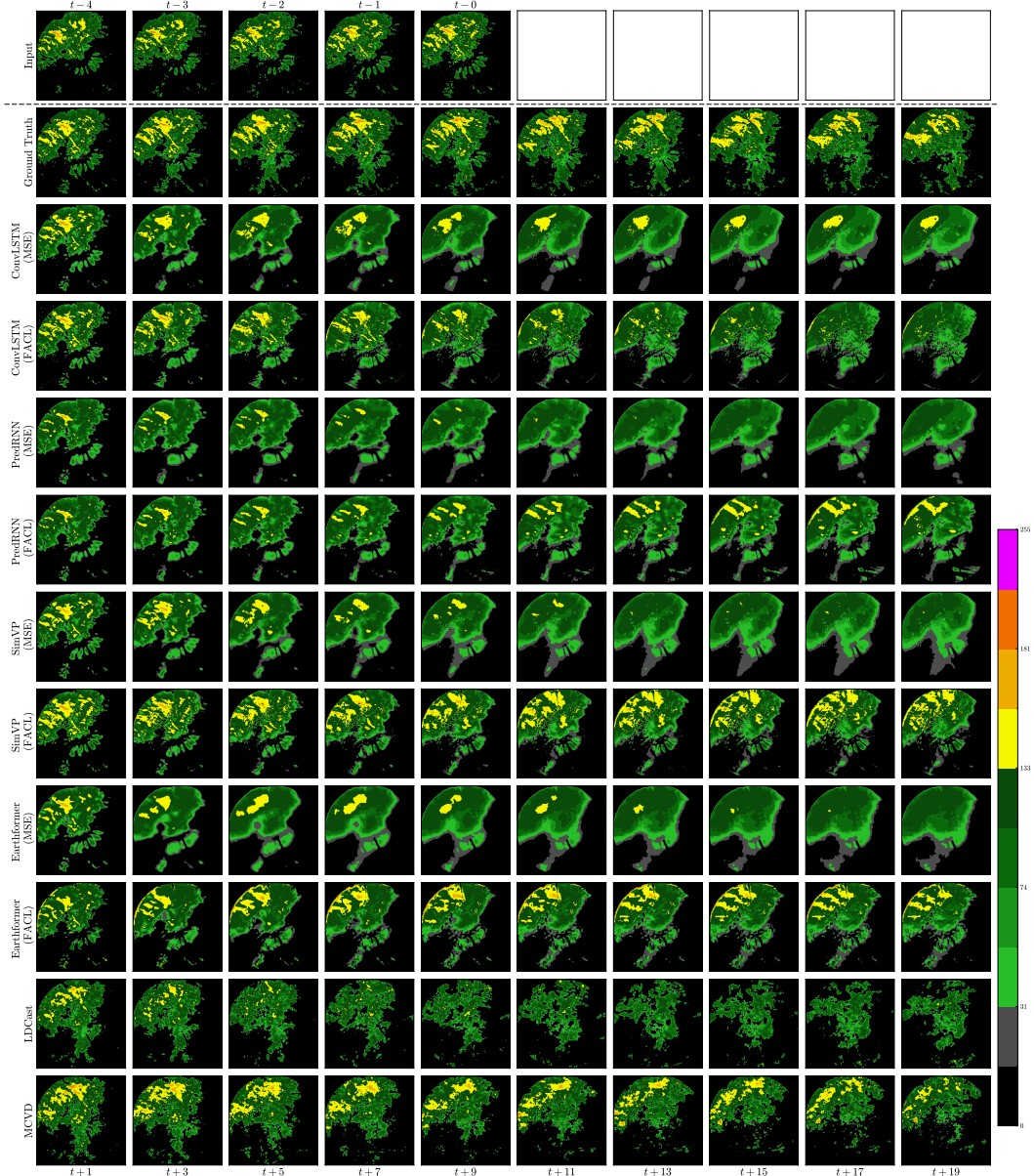

Figure 14: Output frames of the experimented model trained with different losses on HKO-7.

