# OpenReview forum: "Fourier Amplitude and Correlation Loss: Beyond Using L2 Loss for Skillful Precipitation Nowcasting"
_NeurIPS.cc/2024/Conference — NeurIPS 2024 poster_

### Official Review · Reviewer_bJuK · 2024-07-09

**Soundness:** 3
**Presentation:** 3
**Contribution:** 3
**Rating:** 6
**Confidence:** 4

**Summary:**

The paper introduces novel loss functions - the Fourier Amplitude Loss (FAL), Fourier Correlation Loss (FCL) and a Regional Histogram Divergence (RHD) - to improve the realism of the predictions of precipitation nowcasting models without the use of generative models. The loss is applied to established precipitation nowcasting methods and is seen to significantly improve the sharpness and overall realism of the predictions.

**Strengths:**

The paper presents a few interesting loss functions, and in particular the RTD is an interesting approach to improving the representation of precipitation quantiles while not overly penalizing small spatial shifts. The paper also shows comparisons of various key metrics with standard model architectures trained with MSE and with the new losses, showing the advantages of the new losses with

**Weaknesses:**

The authors sensibly use MSE loss as a baseline when evaluating their model. However, there is quite a long history of developing new losses for making precipitation nowcasting produce sharper images and improve the representation of extreme precipitation. As such, I would have wanted to see a paper introducing a new loss function do some comparisons to earlier attempts to improve on the MSE loss if it wants to claim state-of-the-art performance. See e.g.:
https://proceedings.neurips.cc/paper/2017/hash/a6db4ed04f1621a119799fd3d7545d3d-Abstract.html
https://agupubs.onlinelibrary.wiley.com/doi/full/10.1029/2019EA000812
https://www.mdpi.com/2073-4433/10/5/244

**Questions:**

You mention that you introduce FCL because the FAL doesn't fully constrain the spatial positions in the image. A seemingly simpler solution would be to use the FAL in combination with MSE. Have you tried this approach and if so, do you see advantages with using FCL instead?

Also, you are using a loss where you randomize between FAL and FCL and the probability of using FAL increases over time. You motivate this (lines 210-214) as it being tricky to find the correct weighting of FAL and FCL for a linear combination, but I don't quite understand, how is this not functionally equivalent to using a linear combination with the weight of FAL increasing over time?

**Limitations:**

The paper discusses some limitations of the approach, but one point that is not mentioned is that unlike generative approaches to producing realistic nowcasting predictions, the model trained with the new loss function cannot produce multiple outputs for the same input, and is thus unable to provide stochastic predictions where multiple outputs from the model are used to characterize the probability distribution of precipitation.

---

> ### Author Rebuttal · Authors · 2024-08-06
>
> We would like to thank the reviewer’s feedback and suggestion. Here is our response to each of the reviewer bJuK's concern.
>
> ---
>
> > The authors sensibly use MSE loss as a baseline when evaluating their model...
>
> Among the papers the reviewer suggested, BMSE [1] performs linear scaling at different pixel ranges, Multisigmoid loss (SSL) [2] preprocesses the images with linear transformations and nonlinear sigmoid function before applying MSE, [3] tested SSIM and MS-SSIM and recommended MSE+SSIM to be the loss function.
>
> Due to limited time and resources, we present the results by adopting these losses with ConvLSTM trained on Stochastic Moving MNIST. For SSL, we follow the paper and pick $i \in (\frac{20}{70}, \frac{30}{70})$ and $c = 20$. The visualization is reported in the appended PDF.
>
> Stochastic Moving MNIST:
> |Model|Loss|MAE|SSIM|LPIPS|FVD|FSS|RHD|
> |---|---|---|---|---|---|---|---|
> |ConvLSTM|MSE|196.42|0.6975|0.2538|451.54|0.6148|1.1504|
> |ConvLSTM|SSL[2]|175.17|0.7553|0.1906|348.18|0.7225|0.9840|
> |ConvLSTM|MSE+SSIM[3]|184.10|0.7488|0.2573|529.71|0.3514|0.7921|
> |ConvLSTM|FACL|180.10|0.7463|0.1092|82.28|0.8172|0.3391|
>
> HKO-7:
> |Model|Loss|MAE|SSIM|LPIPS|FVD|CSI-m|CSI4-m|CSI16-m|FSS|RHD|
> |---|---|---|---|---|---|---|---|---|---|---|
> |ConvLSTM|MSE|30.43|0.6664|0.3057|791.3|0.2772|0.2282|0.1702|0.2653|1.2453
> |ConvLSTM|BMSE[1]|45.03|0.5537|0.3804|901.9|0.3484|0.3670|0.3354|0.3999|1.7918
> |ConvLSTM|FACL|29.72|0.7168|0.2962|569.1|0.3054|0.3040|0.3351|0.7916|0.4045|0.7916
>
> **We would like to direct the reviewer to view Figures 2 and 3 in the global rebuttal PDF.** We conclude the following summary from the above tables and appended figures, together with some of our previous experience:
> 1. Weighted MSE (such as BMSE) only “tilts” the focus. BMSE severely over-predicts in exchange for an improvement in CSI, sacrificing all other metrics such as MAE, SSIM, LPIPS, FVD, FSS and RHD.
> 2. SSL improves the model performance in general, but still cannot generate clear output under stochastic motion.
> 3. Losses integrating SSIM (and also L1) “dissolves” the prediction to zero over time under uncertainty. Such effect is especially significant for weaker models.
>
> ---
>
> > You mention that you introduce FCL because the FAL doesn't fully constrain the spatial positions in the image. A seemingly simpler solution would be to use the FAL in combination with MSE. Have you tried this approach and if so, do you see advantages with using FCL instead?
>
> Our research indeed started with FAL in combination with MSE in the early stage. We replaced MSE with FCL later due to a few concerns.
>
> **Theoretical concern**:
>
> In Appendix D, we showed that FAL can be broken down into three terms: $L_2(X, \hat{X})$ (that is, MSE in the **image space**), $\Sigma 2X\hat{X}$, and $-\Sigma 2F\hat{F}$. The difference $\Sigma 2X\hat{X} - \Sigma 2F\hat{F}$ corresponds to the translation invariant factor and the L2 term corresponds to the correctness of values. When the image is translated (Figure 4 in paper), the two terms almost cancel out, resulting in a perfect invariance to translation.
>
> If we form the new loss as a linear combination of MSE and FAL, the weighting of the L2 term increases, breaking the balance between the two terms. With this intuition, we believe the resulting loss will not generate better results if we mix MSE with FAL.
>
> **Practical concern**:
>
> Despite the theoretical concern, we did test FAL + MSE. Besides, we also tested FAL + SSIM loss since SSIM is also considered a substitute for MSE.
>
> The models are trained on SEVIR:
> |Model|Loss|MAE|SSIM|LPIPS|FVD|CSI-m|CSI4-m|CSI16-m|FSS|RHD|
> |-|-|-|-|-|-|-|-|-|-|-|
> |ConvLSTM|FAL+MSE|33.3|0.7243|0.3611|351.6|0.3391|0.3728|0.4696|0.3463|1.4143|
> |ConvLSTM|FAL+SSIM|36.7|0.7247|0.3762|395.3|0.3125|0.3433|0.4405|0.3150|1.6056|
> |ConvLSTM|FACL|27.6|0.7402|0.3492|281.8|0.3633|0.3915|0.4838|0.4813|1.3445|
>
> Empirically, FAL+MSE and FAL+SSIM work but FACL outperforms them by some margin. We will append this information to the paper if the reviewer finds it helpful.
>
> ---
>
> >  Also, you are using a loss where you randomize between FAL and FCL ... how is this not functionally equivalent to using a linear combination with the weight of FAL increasing over time?
>
> A linear combination of FAL and FCL with increasing weighting on FAL is functionally equivalent with the current FACL, but the hyper-parameter tuning is different. We attempted to formulate $FACL = w(t) \text{FAL}(X, \hat{X}) + \text{FCL}(X, \hat{X})$ instead of Equation (6). Choosing a good $w(t)$ includes its curve type and maximum value, which can be challenging. Since FFT is orthonormalized (normalized by the image size $\sqrt{HW}$), we set the maximum value of $w(t)$ to $HW$ to generalize FACL to different datasets and image sizes. For HKO-7, $HW=230400$, which may cause numerical instability when multiplying to the FAL term. To avoid the hyper-parameters going too large, we find switching to a linear decay of selection probability between [0, 1] results in a more stable and neat solution. Since the adjustable range of $\alpha$ is small, it is also easier to control the tradeoff between accuracy and realisticity.
>
> ---
>
> > limitation
>
> We think the suggested limitation is built on the premise that the model itself cannot generate multiple forecasts. For example, if a ConvLSTM trained with MSE cannot have ensemble outputs, then the same model trained with FACL cannot do it either. However, if another model trained with MSE can generate diverse outputs, replacing MSE with FACL here can also yield multiple sharper outputs.

---

### Official Review · Reviewer_7mGa · 2024-07-11

**Soundness:** 3
**Presentation:** 2
**Contribution:** 3
**Rating:** 5
**Confidence:** 4

**Summary:**

This paper propose a new Fourier Amplitude and Correlation Loss (FACL) to replace the traditional L_2 losses in precipitation nowcasting task. They evaluated the FACL on one synthetic dataset and three radar echo datasets, which demonstrates their method improve perceptual metrics and meteorology skill scores. Besides, they propose Regional Histogram Divergence (RHD) to improve the error margin in meteorological skill scores.

**Strengths:**

1. The FACL increases the quality of predictions compared to MSE. The model trained with FACL has more small patterns than MSE model, especially for longer lead time. Besides, FACL doesn’t import additional time cost according to Appendix G.
2. Regional Histogram Divergence (RHD), a variation of FSS, replaces the “hit” and “miss” of FSS with regional pixel distribution. It cancels the need of choosing threshold for FSS, alleviating the effect of threshold when evaluating models.

**Weaknesses:**

1. The evaluation is incomplete and unconvincing. The csi-m of EarthFormer is much lower than the results in [1] (0.3982/0.44). Besides, it could include more methods, such as NowcastNet[1] and DiffCast[2].
2. In figure9, compared to generative methods such as LDCast and MCVD, the FACL methods still lack predictions for extreme value and the distribution of predictions given by FACL is slightly different to the ground truth.
3. The intuition of FACL has no strong connection with precipitation nowcasting.
4. The coefficients of FAL and FCL needs to be selected manually, which may hinder the applications of FACL in other datasets and tasks.

Minor:
Models adopt the “prediction and refine” may help improve the related work section.

[1] Skillful nowcasting of extreme precipitation with NowcastNet
[2] Cascast: Skillful high-resolution precipitation nowcasting via cascaded modelling

**Questions:**

Why does the author measure the CSI of SEVIR and MeteoNet with the same threshold? The data in SEVIR are VIL images, while the ones in MeteoNet are radar reflectivity (dbz).

**Limitations:**

The evaluation is incomplete and unconvincing;lack predictions for extreme value;The intuition of FACL has no strong connection with precipitation nowcasting

---

> ### Author Rebuttal · Authors · 2024-08-06
>
> We would like to thank the reviewer’s feedback and suggestions. Here is our response to each of the reviewer 7mGa's concerns.
>
> ---
>
> > The evaluation is incomplete and unconvincing. The csi-m of EarthFormer is much lower than the results in [1] (0.3982/0.44).
>
> We adopted the official Earthfomer model and SEVIR dataloader. However, the training script is re-implemented to ensure a fair test (by only replacing MSE loss with FACL while keeping other factors constant). This causes us to change multiple hyper-parameters, potentially changing the model performance. For example, due to the scheduling for FACL random selection, we should not apply early-stop to the models. Due to resource limitations, the batch size is set to 32 instead of 64, etc. These tiny factors may aggregate and change the overall performance. For example, in DiffCast, their 5-in-20-out setting in SEVIR causes Earthformer to have a CSI-m of **0.2513**, showing that these factors can result in a huge difference in the ultimate performance.
>
> Despite the difference, we do not think it invalidates our evaluation results. Our evaluation does not intend to convince the readers that an arbitrary model with FACL peaks any SOTA model in their optimal state. Instead, with a set of fair tests, we show that a model with FACL is better than the same model with MSE in precipitation nowcasting by its forecasts being more realistic (better LPIPS, FVD) and more skillful (better CSI, FSS, RHD). Such observation is consistently reflected by multiple tables and figures in the paper, both quantitatively and qualitatively. We hope the reviewer can at least agree with such a conclusion.
>
> ---
>
> > Besides, it could include more methods, such as NowcastNet[1] and DiffCast[2].
>
> DiffCast natively does not support 13-in-12-out sequences. Moreover, since 384x384 is too computationally heavy for inference, DiffCast downsamples the SEVIR dataset to 128x128. This causes unfair evaluation especially for LPIPS and pooled CSI which are sensitive to the image size.
> We adopted the official checkpoint of DiffCast and upsample its forecasts to 384x384 on SEVIR:
> |Model|Loss|MAE|SSIM|LPIPS|FVD|CSI-m|CSI_4-m|CSI_16-m|FSS|RHD|
> |-|-|-|-|-|-|-|-|-|-|-|
> |DiffCast(PhyDNet)|-|31.25|0.7568|0.3505|221.0|0.3588|0.3892|0.4874|0.5074|1.4270|
>
> To sum up, DiffCast only has slightly better LPIPS and FVD, while other VP models with FACL in general have better pixel-wise accuracy and skill scores. (The performance of the other models can be found in Table 2 in the paper; the visualizations can be found in Figure 4 in the global rebuttal PDF.)
>
> Again, we would like to reiterate that our core argument is that VP models trained with FACL achieve more realistic and skillful forecasts than those with MSE. The inclusion of generative models serves as a reference to the readers on how well generative models perform under the same setting. We believe further including more references has limited help in delivering our core argument.
>
> ---
>
> > In figure9, ..., the FACL methods still lack predictions for extreme value...
>
> To address the reviewer’s concern, let us present the individual scores with high thresholds: (160, 181, 219) for different pooling sizes 1, 4, and 16 on SEVIR.
>
> |Model|Loss|CSI-160|CSI-181|CSI-219|CSI_4-160|CSI_4-181|CSI_4-219|CSI_16-160|CSI_16-181|CSI_16-219|
> |-|-|-|-|-|-|-|-|-|-|-|
> |ConvLSTM|FACL|0.2755|0.2418|0.1166|0.3082|0.2735|0.1522|0.4110|0.3682|0.2324|
> |SimVP|FACL|0.2848|0.2522|0.1528|0.3133|0.2802|0.1791|0.4188|0.3706|0.2530|
> |LDCast|-|0.1772|0.1460|0.0797|0.2210|0.1842|0.1063|0.3511|0.2986|0.1870|
> |MCVD|-|0.2366|0.2032|0.1174|0.2807|0.2443|0.1504|0.4161|0.3692|0.2579|
>
> From the table, FACL still exhibits better performance in most thresholds, except under a large pooling size. Although generative models attempt to generate more extreme values, the predicted points are distant from the actual location, forming false positives and hence lowering the CSI score. On the other hand, models trained with FACL, despite not predicting extreme values a lot, hit when they do. After considering everything in the intersection-over-union fashion, FACL can usually outperform the reference generative models in predicting extreme values unless with exceptionally large allowance.
>
> ---
>
> > The intuition of FACL has no strong connection with precipitation nowcasting.
>
> The intuition of FACL has the following connection with precipitation nowcasting:
>
> (1) FACL is proposed to solve **blurriness caused by (random) motion**. Previous video prediction models keeps improving on deterministic data by looking into pixel-wise metrics. FACL utilizes the translation invariant property of FAL by treating precipitation events as stochastic data.
>
> (2) FACL is proposed to work on **signal-based data**. We assume there is no cohesiveness between channels such as RGB raw image and medium-range atmospheric forecasting, most preferably having only 1 channel with its range bounded. Among the data types, radar reflectivity and its derivation products such as VIL are good candidates with such properties.
>
> ---
>
> > The coefficients of FAL and FCL needs to be selected manually...
>
> We refer the reviewer to the extra experiments for reviewer w5PB, which shows $\alpha$ is consistently stable in the suggested range (0.1 - 0.4).
>
> ---
>
> > Minor: Models adopt the “prediction and refine” may help improve the related work section.
>
> Thank you for the suggestion. We will include a few more recent models such as CasCast and DGDM to extend the introduction to the “prediction and refine” type of work.
>
> ---
>
> > CSI thresholds for MeteoNet
>
> Thank you for the feedback. The overall idea of choosing the CSI threshold was to select values that span the low, medium, and high range, so we conveniently followed the set used by SEVIR. We will update the CSI thresholds for MeteoNet following the DiffCast paper, by setting them to (12, 18, 24, 32) over 70 dbZ. A preview is reported in the comment.

---

> ### Author Response · Authors · 2024-08-06
> **The updated table for MeteoNet with new CSI thresholds (12, 18, 24, 32)**
>
> Here we present the updated table for MeteoNet with new CSI thresholds (12, 18, 24, 32) based on the reviewer's feedback. Models trained with FACL still largely outperform those trained with MSE. With the increase of the CSI pooling radius, the performance of the diffusion models also drastically increases.
>
> | Model          | Loss  | CSI-m  | CSI_4-m | CSI_16-m |
> | ---  | --- | --- | --- | --- |
> | ConvLSTM  | MSE  | 0.4388 | 0.3989    | 0.3904      |
> | ConvLSTM  | FACL | 0.4161 | 0.4876    | 0.6041      |
> | SimVP         | MSE  | 0.4221 | 0.3748    | 0.3627      |
> | SimVP         | FACL | 0.4008 | 0.4513    | 0.5772      |
> | Earthformer | MSE  | 0.4004 | 0.3327 | 0.2946 |
> | Earthformer | FACL | 0.3594 | 0.4038 | 0.5250 |
> | LDcast         |  -        | 0.2353 | 0.3188   |  0.4804      |
> | MCVD          | -         | 0.3645  | 0.4559 | 0.6148       |

---

> > ### Comment · Reviewer_7mGa · 2024-08-12
> > **The updated commonts**
> >
> > Thanks the authors for the detailed responses. I agree that the proposed FACL is a good alternative to MSE, but I still want to know if the proposed FACL can improve the performance of the SOTA model or what are the benefits of applying it to the sota model

---

> > > ### Author Response · Authors · 2024-08-13
> > > **Regarding new SOTA models**
> > >
> > > We thank the reviewer for the follow-up responses. FACL is proposed to enforce sharpened forecasts for **video prediction** (VP; in contrast to video generation) models. The models we can apply FACL to are mostly **SOTA video prediction models**. The 3 most highlighted models in the paper: ConvLSTM, SimVP and Earthformer, correspond to three types of common model architectures in learning the spatiotemporal patterns: RNNs, CNNs and ViTs. In this category, we are able to provide results on more SOTA models / strong models, such as SimVP v2.
> > >
> > > Stochastic Moving-MNIST:
> > > | Model | Loss | MAE | SSIM | LPIPS | FSS | RHD |
> > > | --- | --- | --- | --- | --- | --- | --- |
> > > |  SimVP v2 (gSTA) | MSE | 171.58 | 0.7625 | 0.1846 | 0.7464 | 0.9479 |
> > > |  SimVP v2 (gSTA) | FACL | 178.55 | 0.7622 | 0.0959 | 0.8210 | 0.3581 |
> > >
> > > The SOTA models the reviewer previously suggested were **video generation models**. In Appendix I, we showed that replacing the MSE reconstruction loss with FACL in GAN/VAE improves different metrics to a considerable degree, but replacing diffusion loss with FACL does not influence the results significantly. Moreover, video generation models usually impose constraints to the data such as the image resolution and sequence length, causing difficulties to align with conventional video prediction models.
> > >
> > > ---
> > >
> > > To sum up,
> > > 1. Applying FACL to **SOTA video prediction models** is the most preferred way of using FACL, which brings huge improvement in both perceptual metrics and skill scores.
> > > 2. Applying FACL to **SOTA VAE/GAN-based models** improves the models.
> > > 3. Applying FACL to **SOTA diffusion-based models** has almost no effect.
> > > 4. We are unable to include some generative models due to lack of implementation source and discrepancy in experimental settings. It is too computationally demanding to train some models (e.g. DiffCast) on the same datasets in full resolution.

---

> > > > ### Comment · Reviewer_7mGa · 2024-08-13
> > > > **The updated commonts**
> > > >
> > > > Thank the authors for their efforts. I will increase my original score. I suggest including the discussion about  SOTA models in the paper, which can help the reader know when to apply the proposed FACL.

---

### Official Review · Reviewer_w5PB · 2024-07-12

**Soundness:** 3
**Presentation:** 3
**Contribution:** 3
**Rating:** 7
**Confidence:** 4

**Summary:**

This paper proposes the FACL loss function and provides theoretical and empiral proofs on how it boosts clarity and structure for images. The paper also shows how the loss behaves with generative setups and additionally proposes a new metric that is tolerant to deformations.

**Strengths:**

- Clearly demonstrates why a naive fourier loss has no benefit over MSE
- Clearly demonstrates how FAL is translation invariant
- Clearly demonstrates why FCL provides global information to predicted pixels instead of pixel-level local information

**Weaknesses:**

- Instead of the stochastic modification to Moving-MNIST, [1] already introduced a chaotic yet deterministic N-Body MNIST to mimic the complexity of Earth system interactions, would have been interesting to see the performance on this dataset
- FVD and LPIPS calculated on the basis of models trained on natural images, difficult to see why it would extend to scientific data

[1] Gao, Z., Shi, X., Wang, H., Zhu, Y., Wang, Y. B., Li, M., & Yeung, D. Y. (2022). Earthformer: Exploring space-time transformers for earth system forecasting. Advances in Neural Information Processing Systems, 35, 25390-25403.

**Questions:**

- Does the alpha parameter swing wildly between datasets? Any good strategy to figure this out?
- What are the benefits of RHD over a simple wasserstein distance between predicted and true distributions or their quantile error?

**Limitations:**

Yes

---

> ### Author Rebuttal · Authors · 2024-08-06
>
> We would like to thank the reviewer’s feedback and suggestion. Here is our response to each of the reviewer w5PB's concern.
>
> ---
>
> > Instead of the stochastic modification to Moving-MNIST, [1] already introduced a chaotic yet deterministic N-Body MNIST to mimic the complexity of Earth system interactions, would have been interesting to see the performance on this dataset.
>
> We are pleased to also present the result of N-Body-MNIST. Due to limited time and resources, we trained ConvLSTM models with MSE and FACL on N-Body-MNIST for 100 epochs, with the results shown below:
>
> | Model | Loss | MAE | SSIM | LPIPS | FVD | FSS | RHD |
> | --- | --- | --- | --- | --- | --- | --- | --- |
> | ConvLSTM | MSE | 57.22 | 0.8946 | 0.1264 | 178.57 | 0.7601 | 0.2301 |
> | ConvLSTM | FACL | 43.11 | 0.9385 | 0.0533 | 80.83 | 0.9198 | 0.1586 |
>
> **For visualization, see Figure 1 in the global rebuttal PDF**
>
> Side notes regarding N-Body-MNIST:
>
> As the reviewer suggested, N-Body-MNIST is deterministic in a closed system. However, we argue that the current research gap (blurry pattern for VP model) is the randomness causing VP models to predict poor quality forecasts. Such randomness cannot be simply solved by “improving the DNN models” as attempted in previous works. Strong models like Earthformer bring huge improvement to N-body-MNIST due to their increase of representation capability. However, upon non-deterministic motion (like Stochastic Moving-MNIST, and actual precipitation events), these strong models’ benefit is limited and they still suffer from blurry predictions. With such intuition, we point the culprit at the MSE loss, which mixes uncertainty into pixel value. By suggesting Stochastic Moving-MNIST as a non-deterministic system, we showed that a tiny stochastic factor (~1 pixel shift per frame) is sufficient to cause severe blurriness for the VP models trained with MSE **regardless of their representation capability** (as visualized in Figure 8 in the paper). FACL, on the other hand, enables the model to learn the expected trajectory and forms a clear output.
>
> ---
>
> > FVD and LPIPS calculated on the basis of models trained on natural images, difficult to see why it would extend to scientific data
>
> Due to the blurriness issue, despite DNN models’ remarkable accuracy, meteorologists find that the model forecasts do not “look like” actual observations. The intuition of using FVD and LPIPS is to show how close the forecasts “look like” actual ones, as a quantified approximation to qualitative studies by humans. For example, the DiffCast [1] paper also reported LPIPS to represent perceptual similarity.
>
> We agree with the reviewer that FVD and LPIPS only do not extend well to scientific data. This is also one of our motivations for proposing RHD, to provide a systematic way of measuring the forecast with a closer approximation to human perception. However, we still include the perceptual metrics to provide readers with a comprehensive study in all three aspects: accuracy (MAE, SSIM), visual realisticity (LPIPS, FVD), and skillfulness (CSI, FSS).
>
> [1] D. Yu, X. Li, Y. Ye, B. Zhang, C. Luo, K. Dai, R. Wang, and X. Chen, “DiffCast: A unified framework via residual diffusion for precipitation nowcasting,” in CVPR, 2024.
>
> ---
>
> > Does the alpha parameter swing wildly between datasets? Any good strategy to figure this out?
>
> We also conducted experiments on $\alpha$ using other datasets such as SEVIR.
>
> | Model | $\alpha$ | MAE | SSIM | LPIPS | FVD | CSI-m | CSI4-m | CSI16-m | FSS | RHD |
> | --- | --- | --- | --- | --- | --- | --- | --- | --- | --- | --- |
> | ConvLSTM | 0.0 | 26.15 | 0.7814 | 0.3502 | 391.37 | 0.4195 | 0.4339 | 0.4710 | 0.5727 | 1.3924 |
> | ConvLSTM | 0.1 | 27.60 | 0.7624 | 0.3508 | 289.49 | 0.3984 | 0.4295 | 0.5073 | 0.5640 | 1.2087 |
> | ConvLSTM | 0.3 | 27.80 | 0.7587 | 0.3312 | 258.24 | 0.3953 | 0.4288 | 0.5242 | 0.5453 | 1.1710 |
> | ConvLSTM | 0.5 | 30.45 | 0.7402 | 0.3492 | 281.82 | 0.3633 | 0.3915 | 0.4838 | 0.4813 | 1.3445 |
>
> Similar to the observation in Table 4, with the increase of $\alpha$ from 0 to 0.3, pixel-wise performance gradually drops and perceptual metrics gradually improve. At around 0.4-0.5, where the model may not fully converge in FCL, the performance starts to decay. Therefore, $\alpha = 0.1$ or $0.2$ as a good default value still holds.
>
> The rule of thumb here is only exposing strong FAL to models that have been fully converged in FCL. (e.g. $\alpha < 0.5$, or have a long training time overall). Once the models converge in FCL, the effect of the proportion of FAL (0.1 - 0.4) is not very significant but a small tradeoff between pixel-wise accuracy and sharpness.
>
> ---
>
> > What are the benefits of RHD over a simple wasserstein distance between predicted and true distributions or their quantile error?
>
> One key component of RHD is to divide the forecast map into smaller patches. With this operation, the measured intensity distribution is limited to a local region, rather than the whole map.
> As for the part of KL divergence, we believe it plays a similar role as Wasserstein distance. In other words, a similar measurement can be achieved if we keep the patching component and replace the KL-divergence with the EM distance.

---

> > ### Comment · Reviewer_w5PB · 2024-08-11
> >
> > Thank you for the thorough response. I particularly appreciate the comments on robustness over the alpha parameters, and am also pleased to see the performance on the additional dataset. I will increase my original score.

---

### Author Rebuttal · Authors · 2024-08-06

To AC and reviewers,

We sincerely appreciate all the constructive reviews from the reviewers. We have summarized the weaknesses the reviewers were concerned the most with, as well as our responses.

---

### Inclusion of more datasets, more previous losses, and more generative models.

Reviewer w5PB suggested also evaluating the models on N-body-MNIST. We performed a simplistic study on N-body-MNIST and the results are similar to that of Stochastic Moving-MNIST, except that more models fail in Stochastic Moving-MNIST due to its random nature. We will also include such a study in the paper.

Reviewer bJuK suggested including other loss functions proposed to improve the model performance. We extended the experiments by also comparing with BMSE, SSL and MSE+SSIM. Although some of them improve the model performance to a certain degree, none could produce a sharp prediction under stochastic data.

Reviewer 7mGa suggested including more generative models such as DiffCast and NowcastNet. We are unable to reproduce NowcastNet promptly due to the lack of implementation of its loss function, but we reported the performance of pre-trained DiffCast under our setting on SEVIR. However, we believe the pool of generative models should not be the most prioritized focus, compared with other aspects such as the generality of FACL and ablation studies.

---

### Picking $\alpha$

Both reviewers w5PB and 7mGa questioned the stability of the choice of $\alpha$. As explained in the rebuttal, $\alpha \in [0.1, 0.4]$ is generally stable. The underlying rule for the selection of $\alpha$ is to ensure the models are fully converged in FCL first before exposure to FAL. As long as the models converge in FCL, the effect of $\alpha$ is not sensitive but a little tradeoff between pixel-wise accuracy and sharpness. Since there is no absolute telling on which type of metrics is superior, we believe the freedom to control the tradeoff should be considered a benefit rather than a limitation.

---

### Other potential formulations for FACL

Reviewer bJuK inquired about other potential formulations for FACL, such as a linear combination of FAL and FCL, and replacing FCL with MSE, etc. We showed that we avoided linear combination due to concerns of bad hyper-parameter causing numerical instability, and our presented formulation of FACL is the most performant among the tested variations.

---

## The appended PDF for figures.

We present 4 additional figures to better illustrate our points to the reviewers.
- To reviewer w5PB: Figure 1 visualizes the output frames of the models on N-body-MNIST.
- To reviewer bJuK: Figures 2 and 3 show the output frames for models with different losses, including BMSE, SSL and MSE+SSIM, trained on Stochastic Moving-MNIST and HKO-7 respectively.
- To reviewer 7mGa: Figure 4 visualizes the output of the DiffCast model for the same test sample as Figure 9 in the paper.

---

### Decision · Program_Chairs · 2024-09-25

**Decision:**

Accept (poster)

**Comment:**

The paper proposes novel loss functions designed to enhance the prediction quality of precipitation nowcasting models without the use of generative models. This non-generative approach notably sharpens the spatiotemporal predictions. All reviewers agree the technical contribution and soundness in the initial pass of reviewing. Also, the authors' rebuttal fully addressed some questions and suggestions raised by reviewers. The AC recommends accepting this paper.